# Effect of voluntary human mobility restrictions on vector-borne diseases during the COVID-19 pandemic in Japan: A descriptive epidemiological study using a national database (2016 to 2021)

Kenji Hibiya[1,2☯]*, Akira Shinzato[1☯], Hiroyoshi Iwata[3], Takeshi Kinjo[1], Masao Tateyama[4], Kazuko Yamamoto[1], Jiro Fujita[1,4]

1 Department of Infectious, Respiratory, and Digestive Medicine, University of the Ryukyus, Nishihsara, Japan, 2 Department of Pathological Diagnosis, University of the Ryukyu Hospital, Nishihara, Japan, 3 Center for Environmental and Health Sciences, Hokkaido University, Sapporo, Japan, 4 Ohama Dai-ichi Hospital, Omoto-kai Group, Naha City, Okinawa, Japan

☯ These authors contributed equally to this work.
* kenjipo@med.u-ryukyu.ac.jp

## Abstract

The coronavirus disease 2019 (COVID-19) pandemic not only encouraged people to practice good hygiene but also caused behavioral inhibitions and resulted reduction in both endemic and imported infectious diseases. However, the changing patterns of vector-borne diseases under human mobility restrictions remain unclear. Hence, we aimed to investigate the impact of transborder and local mobility restrictions on vector-borne diseases through a descriptive epidemiological study. The analysis was conducted using data from the National Epidemiological Surveillance of Infectious Diseases system in Japan. We defined the pre-pandemic period as the period between the 1st week of 2016 to the 52nd week of 2019 and defined the pandemic period as from the 1st week of 2020 to the 52nd week of 2021, with the assumption that human mobility was limited throughout the pandemic period. This study addressed 24 diseases among notifiable vector borne diseases. Datasets were obtained from weekly reports from the National Epidemiological Surveillance of Infectious Diseases, and the incidence of each vector-borne disease was examined. Interrupted time series analysis was conducted on the epidemic curves for the two periods. Between the pre- and post-pandemic periods, the incidence of dengue fever and malaria significantly decreased, which may be related to limited human transboundary mobility ($p$ = 0.003/0.002). The incidence of severe fever with thrombocytopenia syndrome, scrub typhus, and Japanese spotted fever did not show changes between the two periods or no association with human mobility. This study suggests that behavioral control may reduce the incidence of new mosquito-borne diseases from endemic areas but may not affect tick-borne disease epidemics within an endemic area.

**Data Availability Statement:** All relevant data are within the manuscript and its Supporting information files.

**Funding:** The authors received no specific funding for this work.

**Competing interests:** The authors have declared that no competing interests exist.

## Introduction

In December 2019, the coronavirus disease 2019 (COVID-19) pandemic began with an outbreak in Wuhan (Hubei Province, People's Republic of China). In Japan, the infection spread through tourists and others returning from China to the country [1, 2]. The domestic outbreak and global pandemic brought about changes in hygiene behaviors that seemed to prevent the spread of droplet infections, such as influenza [3]. The COVID-19 pandemic has also led to restrictions on immigration and mass gatherings and the implementation of curfews [4]. For example, worldwide, the passenger volume on international flights decreased by 63.2% in 2020 compared with that in 2019 [5]. In Japan, restrictions were placed on entry from endemic countries, and during the nationwide declaration of a state of emergency, restrictions were placed on movement out of prefectures. These restrictions effectively slow the onset and spread of influenza virus epidemics in new (non-endemic) areas [6]. However, whether mobility restriction is effective against epidemics of vector-borne diseases (VBDs) remains unclear. Forced restriction of human mobility certainly seemed to have reduced the incidence of VBDs [7]. Non-mandatory local mobility restrictions implemented in Taiwan and Japan may exhibit different effects with regard to domestic VBDs epidemics.

Today, the advent of larger vessels and aircrafts has enabled many people and goods to move across borders at an unprecedented rate. The introduction and transmission of malaria and dengue fever to non-endemic regions have been reported via air travel [8]; the aforementioned means of transport are also thought to be directly responsible for the rapid spread of disease vectors. It has been noted that vectors are often transported from endemic to non-endemic areas along with sea cargo shipment of commodities; if the environment in these locations is suitable for the disease vectors, transmission can spread to these non-endemic areas [9]. It is, therefore, also important to consider commodity transport in the control of VBDs. Nevertheless, it is unclear whether the increase or decrease in foreign and domestic trade has an effect on the incidence of domestic VBDs.

It is known that the regional incidence of VBDs often depends on the habitat range of the vector. However, it is not clear whether these differences in the regional incidence are affected by changes in human mobility due to the pandemic of COVID-19. Since the incidence of malaria and dengue fever, in which humans are the source of infection, may correlate with human mobility, a reduction in the number of travelers from endemic areas may alter the suspected transmission area [7, 8]. In addition, several tick-borne diseases are rarely transmitted by humans, so the regional differences may not change.

Tick-borne pathogens are maintained in enzootic cycles involving ticks and wild animal hosts, with epizootic spread to other mammals, including livestock and humans [10]. Therefore, natural habitats of animals, especially wild beasts and birds, may harbor many ticks. On the other hand, malaria and dengue fever are diseases transmitted by mosquitoes to humans/animals. Pathogen-bearing mosquitoes can be found around humans dwelling in urban areas [11]. Therefore, the incidence of VBDs may be affected by the movement of people in urban as well as rural areas.

It is unclear whether the sex- and age-specific incidence rates of VBDs changed during the COVID-19 pandemic. A study in Taiwan on scrub typhus considered the association between age groups and specific occupations, such as farmer or soldier [12]. Furthermore, the female population in China has a reportedly higher incidence of SFTS [13], which may be due to their greater exposure to ticks during agricultural work and susceptibility to the disease after infection [13]. As a high proportion of agricultural work is known to be the work at the time of infection with VBDs in Japan [14], the sex and age distributions of affected individuals may be influenced by the occupation.

This study aimed to examine the impact of human mobility and freight transportation restrictions on the incidence of VBDs in Japan during the COVID-19 pandemic. The list of the research questions and the related topics in the present study is presented below:

1. Relationship between the incidences of VBDs and human mobility

   a. Access to healthcare

   b. Domestic passenger transportation activity

   c. Foreign arrival

   d. Returnees

   e. Domestic human mobility: Long-distance travel / Population staying in local or national parks

2. Relationship between the incidences of VBDs and commodity distribution

   a. Foreign trade

   b. Domestic trade

3. Relationship between the incidences of VBDs and sex and age

4. Other relevant factors with the incidence of VBDs

   a. COVID-19 epidemic status in Japan

   b. Geographical factor: World region / domestic region

## Materials and methods

### Study design and period

A descriptive epidemiological study design was adopted to explore the effects of restrictions on the movement of people and commodities on the incidence of various VBDs during the COVID-19 pandemic. The survey period was defined as the period from January 2016 to December 2021 and was divided as follows: the pre-pandemic (2016–2019) and pandemic (2020–2021) periods. We set the study period to begin in 2016 because the number of domestic Chikungunya fever cases was available from 2016 in Japan. The pandemic period was set until 2021 because various confounding factors were considered from 2022 owing to the relaxation of behavioral restrictions, holding of events, and reopening of international flights. We examined the activity of the tourism industry and passenger transport services from January 2019 to September 2022 during the COVID-19 pandemic in Japan (S1 Fig). Data were obtained from monthly reports of Indices of Tertiary Industry Activity released by the Ministry of Economy, Trade, and Industry [15] (S1 Table). It showed that from 2020 to 2021, people's mobility was somewhat restrained. However, since the activities of passenger transport services had increased since the spring of 2022, we assumed that people's activity increased. Therefore, we assumed that some level of stability was maintained from 2020 to 2021. In addition, the declaration of a state of emergency was the highest behavioral restriction to prevent the spread of COVID-19 during the pandemic period in Japan (S1 Fig). The national and local governments requested residents to refrain from cross-border travel and avoid unnecessary trips outside during the state of emergency [16]. The quasi-emergency measure was the second major restriction after the declaration of a state of emergency (S1 Fig). In Japan, a significant reduction in inter-prefectural travel was achieved even

without the major restriction by the government between the pandemic periods [17]. We confirmed the appropriateness of the present study period.

## Data collection and primary measures

**COVID-19 cases.** We obtained open data on COVID-19 cases that occurred between January 16, 2020 and December 31, 2021 from the Ministry of Health, Labor and Welfare (MHLW) [18]. Data on the daily incidence of new positive cases who had undergone polymerase chain reaction tests for SARS-CoV-2 or antigen testing for SARS-CoV-2 were used in this study. These domestic cases did not include cases of airport quarantine. Data are presented in the S1 Table.

**Foreign arrivals.** Foreign arrivals were examined to understand how foreign travel impacts the incidence of VBDs. The number of foreign nationals who entered Japan between January 2016 and December 2021 was examined. Data regarding the number of immigrants were obtained from reports issued monthly by the Immigration Service Agency of Japan [19]. Data are presented in the S2 Table.

**Returnees.** Returnees were examined to understand how foreign travel impacts the incidence of VBDs. The number of returnees to Japan between January 2016 and December 2021 was examined based on statistical data on emigration and immigration management that were issued monthly by the Immigration Service Agency of Japan [19]. Data are presented in the S2 Table.

**Cargo containers handled in foreign trade.** Cargo containers were examined to understand how overseas commodity distribution impacts the incidence of VBDs The number of cargo containers handled in foreign trade was examined from January 2016 to December 2021 using datasets released by the Harbor Modernization Promotion Committee [20] (S3 Table). In this study, foreign trade by ship was considered as the transportation mode of interest with regard to freight transport. More specifically, in Japan (which is surrounded by sea), goods are moved overseas either by air or by ship. However, airplanes are often used for transporting lightweight and expensive items (such as personal computers and luxury brand goods), whereas ships are mainly used for daily necessities, fuel, and industrial raw materials and can be used to transport large quantities. In Japan, ships currently account for 99.6% of trade by weight [21].

**Domestic transport volume by transportation service.** Domestic transport volumes were examined to determine the extent to which the long-distance movement of people and goods within a country affects the incidence of VBDs. The number of passengers and the cargo volume for motor vehicles from 2016 to 2021 was examined based on a survey on motor vehicle transport data conducted by the Ministry of Land, Infrastructure, Transport and Tourism (MLIT). The number of passengers and cargo volume undertaken by railway transportation from 2016 to 2021 was examined based on the annual report of railway transportation statistics released by the MLIT [22]. The number of railway passengers accounted only for people who boarded the bullet train, excluding commuter pass users. This is because most commuter pass users were unlikely to visit vector habitats. The bullet train is the most popular train for long-distance travel in Japan. The number of passengers and cargo volume undertaken by coastal vessels from 2016 to 2021 was examined based on a statistical survey on coastal vessel transport conducted by the MLIT [22]. The number of passengers and cargo volume undertaken by domestic regular flights from 2016 to 2021 was examined based on data from the air transport statistics survey conducted by the MLIT [22]. Those data are presented in the S4 Table.

Passenger-kilometers and freight tonne-kilometers were calculated according to the number of passengers or the cargo volume (S4 Table). Passenger-kilometers represent the

cumulative total of the number of individual passengers (persons) transported multiplied by the distance (kilometer) traveled by each passenger. The relevant formula is "passenger-kilometer = number of passengers transported × distance traveled;" and this is an important indicator of the scale of transportation. The transported tonne-kilometer represents the accumulation of individual cargoes transported (in tons) multiplied by the distance (in kilometers) each cargo was transported. The formula is "ton-kilometer transported = tonnage transported × distance transported;" and this is an important indicator of the scale of cargo transport. We intended that this study focused on long-distance travel for sightseeing and outdoor activities, rather than travel within the daily living area. Therefore, we employed the amount of passenger kilometers rather than passenger number. The ton-kilometer was chosen to match passenger kilometers.

**Domestic human mobility trends.** Domestic human mobility was examined to understand the extent to which people's mobility in the country affected the incidence of VBDs. We obtained the data sets (February 2020 to October 2022) from the COVID-19 Community Mobility Reports provided by Google [23]. These reports showed how the number of visitors to parks/outdoor spaces has changed compared to baseline days (averages from the 5-week period from January 3, to February 6, 2020. The locations where people visited were classified into several categories. We selected parks or outdoor spaces and residential areas as locations. The reason for this selection was that parks/outdoor spaces are considered one of the most likely places to be bitten by ticks. Residential areas were also used as an indicator for staying indoors. The "parks" included places like local parks, national parks, public beaches, marinas, dog parks, plazas, and public gardens. The "residential areas" were defined as distances within a radius of 2 km from the home. The data are presented in the S5 Table.

**Datasets on access to healthcare.** The frequency of access to healthcare was examined to understand whether underestimation may have biased findings. We collected data on the number of health insurance claims from monthly statistical reports as surrogate indicator of access to healthcare. The data collected from the monthly statistical reports were published by the Health insurance claims review & Reimbursement Services [24]. Those data are presented in the S6 Table.

**Datasets on vector-borne diseases.** In this study, VBDs included all vector-borne diseases among the nationally notifiable diseases within the National Epidemiological Surveillance of Infectious Diseases (NESID) system operated by the National Institute of Infectious Diseases and the MLHW. The reported number of infectious disease cases was collected from the Infectious Diseases Weekly Reports published by the National Institute of Infectious Diseases [25]. The epidemiological weekly data obtained were converted into monthly data according to the Weeks Ending Log [26]. The data are presented in the S7 Table.

Data on incidence categorized by suspected region of infection, age group, and sex were also obtained [27]. These information obtained was queried in standard physician interviews. As for the estimated region of infection, physicians extrapolated these details from the patient's travel history, location of the insect bite, and other details.

The annual incidence of each disease, classified according to whether the presumed area of infection was national or international, is shown in S8 Table. Estimated world regions of infection for diseases with a high number of imported cases were categorized based on the United Nations classification of global regions (see https://unstats.un.org/unsd/methodology/m49/). The data is presented in the S9 Table. For diseases that are endemic in Japan, the estimated transmission areas were divided based on the regional divisions of the country. Japan was divided into three regions as follows: Hokkaido, Eastern Japan, and Western Japan. Hokkaido was distinguished from Eastern Japan by the Blakiston Line, a biological boundary between Eastern and Western Japan (Fig 1). Eastern and Western Japan were additionally separated by

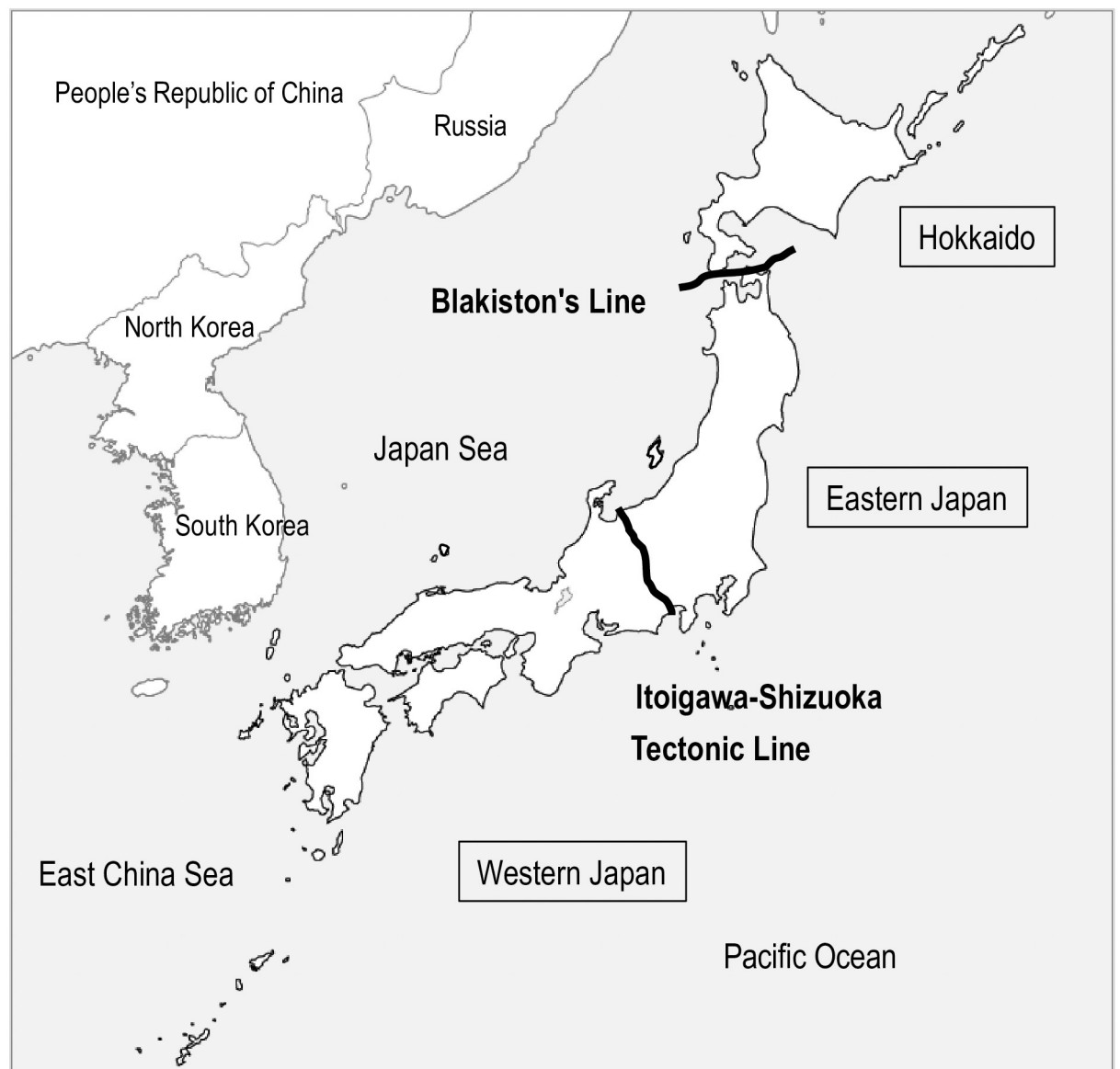

**Fig 1. Location of the Blakiston and Itoigawa–Shizuoka Tectonic Lines in the Japanese archipelago and the regional division of Japan based on these lines.**

the Itoigawa–Shizuoka Tectonic Line, a major fault line and a biological boundary (Fig 1). The data classified by geographic region are presented in the S9 Table. Data on the number of domestic VBDs stratified by age and sex are shown in the S10 Table. The figures were corrected for the proportion of each age group and sex distribution in the population census [28]. The patient characteristics for 2021 were not available and were thus excluded from this analysis.

The area north of the Blakiston Line was defined as Hokkaido, and the area south of the Blakiston Line was divided into Eastern and Western Japan in reference to the Itoigawa–Shizuoka Tectonic Line. This blank map was created by processing an electronic topographic map 25000 (Geospatial Information Authority of Japan) [29].

## Statistical analyses

Pearson correlation was used to examine the relationship between entrants to the country with monthly incidences of VBDs or between domestic human mobility obtained from Google mobility data sets with monthly incidences of VBDs from January 2016 to October 2022.

Interrupted time series analysis (ITSA) was conducted using monthly data to determine significant changes in the incidences of individual diseases during the pre- and post-COVID-19 pandemic following the method described by Linden [30]. ITSA is a method of evaluating the impact of an intervention on an outcome (the causal effect) in terms of the level of change (the change in intercept) or the trend in change (the change in slope) within a time series [30]. The analysis was focused on diseases whose monthly incidence could be statistically analyzed (an average monthly incidence of >1.5) or whose international importance was high, such as dengue fever, malaria, sever fever with thrombocytopenia syndrome (SFTS), scrub typhus, and Japanese spotted fever (JSF). First, we examined whether there was a linear underlying time trend of the incidences of VBDs during the pre-pandemic and pandemic periods using the Cochran–Armitage test. Ordinary least-squares regression models designed to adjust for autocorrelation were employed for the single-group ITSA, which included data on the pre-pandemic and pandemic periods.

All statistical analyses were conducted using Light Stone®STATA®statistical software (ver.15,StataCorp LP, College Station, TX, USA).

## Results

### COVID-19 incidence and access to healthcare

We initially investigated the number of confirmed cases of COVID-19 in Japan. Peaks of the pandemic were observed in April and August in 2020 and January, May, and August in 2021 (S1 Fig). For each outbreak, the government declared a state of emergency and recommended that people limit their movement. Moreover, after the first outbreak was over, the Government implemented a travel assistance program to boost the local economy. Subsequently, the travel and transport industry index showed a temporary increase. We examined restrictions in access to healthcare during the COVID-19 pandemic (S2 Fig). S2 Fig shows the monthly number of patients who visited the clinic (19 beds or less) as outpatients with health insurance. The value shows the percentage change in the same month in 2019 as 100. The departments concerned with the examination of VBDs were internal medicine, pediatrics, and dermatology. The total number of outpatients was at its lowest from April to May 2020 and in January 2021, when the state of emergency was declared; however, there was a rapid recovery trend the following month. The dynamics differed by department. Pediatrics and internal medicine showed large declines, whereas dermatology showed a smaller decline.

### International and domestic movements

The number of foreigners entering Japan had been on the rise prior to when the COVID-19 pandemic began (S3a Fig). In December 2019, the proportion of Asian foreign nationals entering the country was 82.8%, whereas that of non-Asians was 17.2%. In February 2020 (at the start of the global COVID-19 pandemic), the number of immigrants began to decline. Although the ratio of visitors from Asia remained high despite this decline, the ratio of visitors from Asia declined, and the ratio of visitors from Europe increased only in July and August in 2021 (S3a Fig inset). Since 2016, the number of Japanese returnees had remained above 1 million each month, although there had been monthly fluctuations (S3b Fig). However, this number has dropped to 10,000 by April 2020, and was less than 5% of the total returnees in 2019.

The number of containerized cargo handled in external trade during the pandemic declined at a rate of only -2.3% during the pandemic compared to the pre-pandemic period (S3c Fig). The mobility of people within the country during the pandemic period showed a decrease within each form of transportation service. Person-kilometers decreased 58.3% by automobile, 63.5% by railway, and 55.1% by air transportation compared to the pre-pandemic period (S4 Table). Trends in domestic cargo during the pandemic period showed a small decrease within each of the transportation services, except aircraft (S4 Table).

## Incidence of vector-borne diseases

The incidence of VBDs among notifiable infectious diseases during pre- and post-pandemic periods in Japan is shown in Table 1. The incidence of Zika, Chikungunya fever, malaria, and dengue fevers, decreased by more than half during the pandemic compared to the pre-pandemic period. On the other hand, the incidence of relapsing fever, SFTS, scrub typhus, JSF, and Lyme disease increased during the pandemic compared to the pre-pandemic period.

The epidemic curves of representative VBDs are shown in S4 Fig. Annual cyclicity with a peak in August to September was observed for dengue fever and malaria (S4a Fig). For both diseases, apparent seasonality was obscured during the pandemic period. SFTS, scrub typhus,

**Table 1. Incidence of vector-borne diseases stratified by the estimated transmission area, 2016–2019 and 2020–2021.**

| Vector-borne diseases | Pre-pandemic period (2016–2019) | | | Pandemic period (2020–2021) | | | % change per year vs. 16–19' |
|---|---|---|---|---|---|---|---|
| | Domestic | Imported | Total (per year) | Domestic | Imported | Total (per year) | |
| Crimean-Congo hemorrhagic fever | 0 | 0 | 0 (0.0) | 0 | 0 | 0 (0.0) | n.r. |
| Plague | 0 | 0 | 0 (0.0) | 0 | 0 | 0 (0.0) | n.r. |
| West Nile fever | 0 | 0 | 0 (0.0) | 0 | 0 | 0 (0.0) | n.r. |
| Yellow fever | 0 | 0 | 0 (0.0) | 0 | 0 | 0 (0.0) | n.r. |
| Omsk hemorrhagic fever | 0 | 0 | 0 (0.0) | 0 | 0 | 0 (0.0) | n.r. |
| Kyasanur forest disease | 0 | 0 | 0 (0.0) | 0 | 0 | 0 (0.0) | n.r. |
| Western equine encephalitis | 0 | 0 | 0(0.0) | 0 | 0 | 0 (0.0) | n.r. |
| Eastern equine encephalitis | 0 | 0 | 0 (0.0) | 0 | 0 | 0 (0.0) | n.r. |
| Venezuelan equine encephalitis | 0 | 0 | 0 (0.0) | 0 | 0 | 0 (0.0) | n.r. |
| Rift Valley fever | 0 | 0 | 0 (0.0) | 0 | 0 | 0(0.0) | n.r. |
| Rocky Mountain spotted fever | 0 | 0 | 0 (0.0) | 0 | 0 | 0 (0.0) | n.r. |
| Zika fever | 0 | 20 | 20 (5.0) | 0 | 1 | 1 (0.5) | -90.0 |
| Chikungunya fever | 0 | 72 | 72 (18.0) | 0 | 3 | 3 (1.5) | -88.0 |
| Dengue fever | 3 | 1,245 | 1,248 (312.0) | 0 | 53 | 53 (26.5) | -88.2 |
| Malaria | 0 | 222 | 222 (55.5) | 0 | 50 | 50 (25.0) | -55.0 |
| Relapsing fever | 27 | 1 | 28 (7.0) | 25 | 0 | 25 (12.5) | +78.6 |
| SFTS | 327 | 0 | 327 (81.8) | 186 | 0 | 186 (93.0) | +13.7 |
| Scrub typhus | 1,797 | 4 | 1,801 (450.3) | 1,047 | 0 | 1,047 (523.5) | +16.3 |
| Tick-borne encephalitis | 4 | 0 | 4 (1.0) | 0 | 0 | 0 (0.0) | -100 |
| Japanese spotted fever | 1,228 | 1 | 1,229 (307.3) | 905 | 1 | 906 (453.0) | +47.4 |
| Japanese encephalitis | 23 | 0 | 23 (5.8) | 8 | 0 | 8 (4.0) | -31.0 |
| Lyme disease | 44 | 13 | 57 (14.3) | 50 | 0 | 50 (25.0) | +74.8 |
| Epidemic typhus | 0 | 0 | 0 (0.0) | 0 | 0 | 0 (0.0) | n.r. |
| Tularemia | 0 | 0 | 0 (0.0) | 0 | 0 | 0 (0.0) | n.r. |

n.r. no case reported. SFTS: severe fever with thrombocytopenia syndrome.

and JSF each showed a clear similarity with the respective epidemic curves of the pre-pandemic period (S4b Fig).

We examined the correlation between the incidence of dengue fever and malaria and the number of new arrivals to the country (Fig 2a). Both diseases showed a positive correlation with the number of foreign arrivals and returnees (dengue fever: 0.604/0.727, Malaria: 0.463/0.543). Fig 2a shows that the incidence of dengue fever was increasing as the number of people entering the country increased. We also examined the correlation between the incidence of representative endemic infectious diseases and people's domestic mobility change (Fig 2b). SFTS and scrub typhus showed less correlation for mobility change in parks (0.215/-0.052). JSF showed a positive correlation with mobility change in parks (0.400). However, those infectious diseases showed periodic fluctuations on an annual basis regardless of the COVID-19 pandemic.

We conducted ITSA, a statistical analysis, to examine significant differences between the pre-pandemic and pandemic periods (Table 2, Fig 3). Before the COVID-19 pandemic began, dengue fever/malaria onset per month did not significantly increase or decrease every month. In the first year of the pandemic (2020), there appeared to be a significant decrease in dengue fever ($p = 0.003$) and malaria ($p = 0.002$) onset per month, followed by a non-significant increase in the annual trend of dengue fever and malaria onset per month. No significant differences were observed for SFTS ($p = 0.265$), scrub typhus ($p = 0.207$), and JSF ($p = 0.514$) during the pre-pandemic and pandemic periods. However, there was a change in slope from 0.021 to 0.016 for SFTS, from 0.115 to 0.511 for scrub typhus, and from 0.062 to 0.268 for JSF.

## Infected areas (global and domestic)

We examined changes in the distribution of suspected regions of infection (world regions) for VBDs between the pre-pandemic and pandemic periods (Table 3). Zika fever was reported in the pre-pandemic period, especially by travelers from Asia and the Americas, at a rate of over 40%; however, in the pandemic period, the number of cases decreased. Chikungunya fever was significantly more prevalent in visitors from Asia (91.7%) during the pre-pandemic period; however, the number of cases decreased during the pandemic period. Dengue fever was the most prevalent among visitors from Asia during the pre-pandemic period and the pandemic periods (87.3% and 81.1%, respectively). Malaria was especially common in visitors from Africa (75.1%) and Asia (14.2%) during the pre-pandemic period. Although these cases decreased during the pandemic period, the ratio was generally maintained (Africa 68.1% vs. Asia 12.8%).

Relapsing fever was confined to Hokkaido during the pandemic, although endemics were seen in areas outside of Hokkaido before the pandemic (Table 3). SFTS and scrub typhus were observed at significant rates in Western Japan and Eastern and Western Japan, respectively (Table 3). JSF was observed at significant rates in Western Japan in both periods, whereas SFTS, scrub typhus, and JSF were not reported in Hokkaido throughout the study period. Japanese encephalitis was observed at significant rates in Western Japan during both periods. During the pre-pandemic period, the incidence of Lyme disease was predominantly observed in Hokkaido (56.1%) but was also observed in Eastern (24.6%) and Western Japan (19.3%). Moreover, during the pandemic, a clear increase in cases in the Hokkaido region (88.0%) and a decrease in cases in Eastern (4.0%) and Western (8.0%) Japan were observed.

## Incidence based on age and sex

We compared the number of new cases of VBDs prevalent in the country based on age between the pre-pandemic and pandemic periods (Table 4). Relative to all infectious diseases,

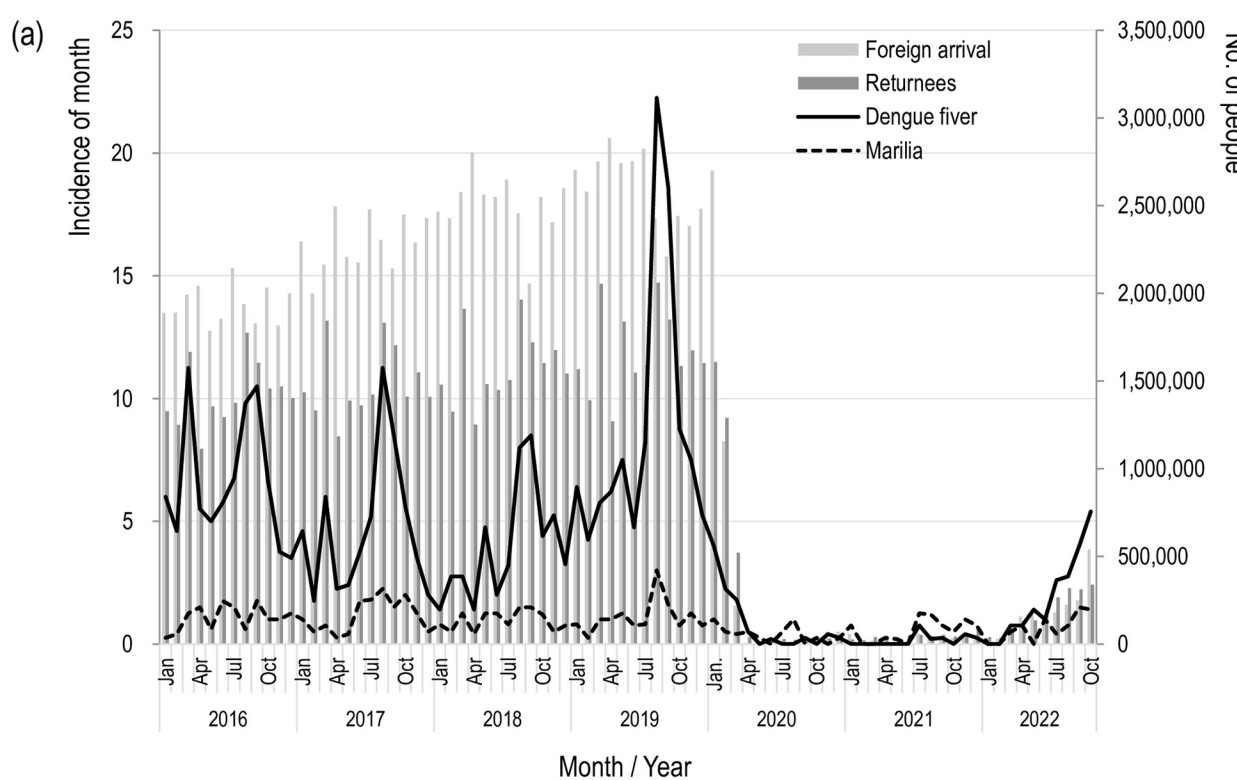

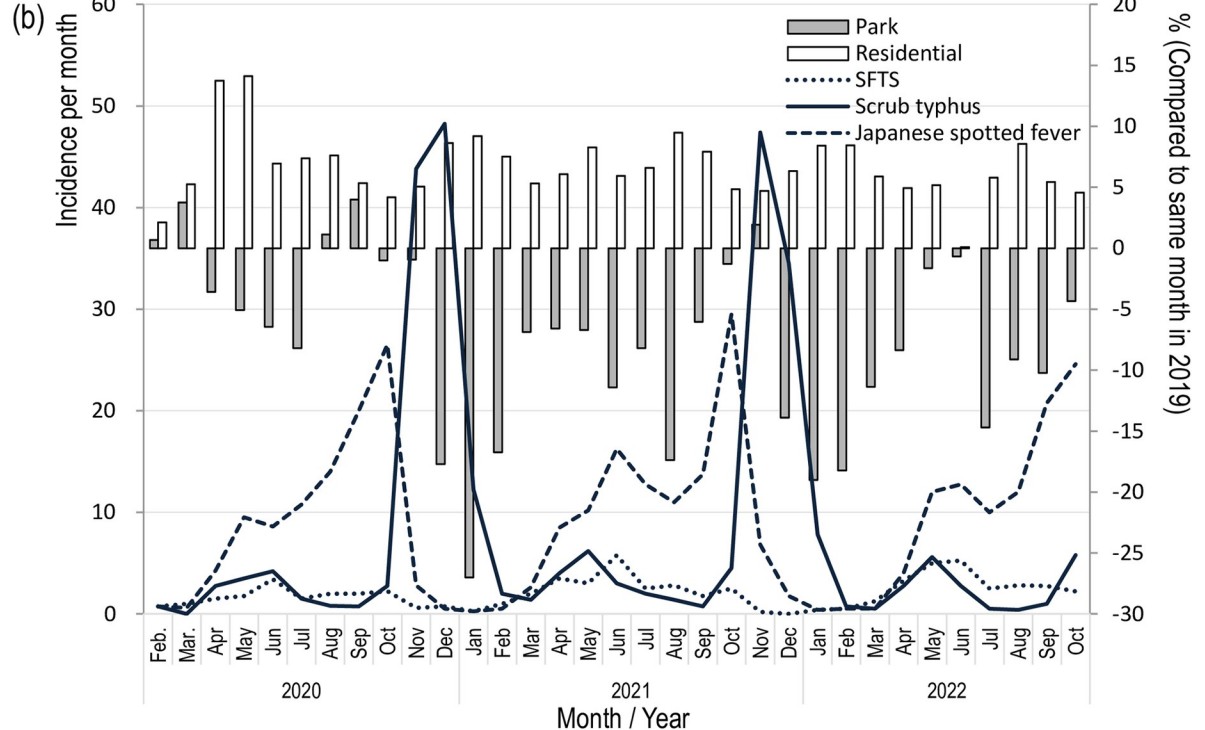

**Fig 2. Relationship change following human mobility change with incidence of vector-borne diseases between January 2021 to October 2022.**
a) Number of foreign arrivals/Japanese returnees and incidence of vector-borne diseases b) Number of visitors in parks and at home and incidence of vector-borne diseases.

**Table 2. Interrupted time-series analysis of the incidences of vector-borne diseases for the pre- and post-pandemic periods.**

| Diseases | | Coefficient | *p*-value | 95% CI |
|---|---|---|---|---|
| Dengue fever | Slope prior to intervention | 0.060 | 0.322 | -0.060 to 0.180 |
| | Immediate change post-intervention | -6.225 | 0.003 | -10.310 to -2.140 |
| | Change in slope post-intervention | -0.128 | 0.076 | -0.270 to 0.014 |
| Malaria | Slope prior to intervention | 0.002 | 0.724 | -0.011 to 0.015 |
| | Immediate change post-intervention | -0.812 | 0.002 | -1.322 to -0.302 |
| | Change in slope post-intervention | 0.011 | 0.452 | -0.017 to 0.039 |
| SFTS | Slope prior to intervention | 0.021 | 0.171 | -0.009 to 0.052 |
| | Immediate change post-intervention | -0.712 | 0.265 | -1.978 to 0.553 |
| | Change in slope post-intervention | 0.016 | 0.745 | -0.084 to 0.116 |
| Scrub typhus | Slope prior to intervention | 0.115 | 0.539 | -0.256 to 0.487 |
| | Immediate change post-intervention | -9.177 | 0.207 | -23.541 to 5.187 |
| | Change in slope post-intervention | 0.511 | 0.327 | -0.522 to 1.545 |
| JSF | Slope prior to intervention | 0.062 | 0.314 | -0.060 to 0.185 |
| | Immediate change post-intervention | -2.373 | 0.514 | -9.594 to 4.849 |
| | Change in slope post-intervention | 0.268 | 0.289 | -0.232 to 0.768 |

CI: confidence interval, JSF: Japanese spotted fever, SFTS: severe fever with thrombocytopenia syndrome.

the proportion of new cases by age did not change significantly between the pre-pandemic and pandemic periods. The incidence of relapsing fever was similar between the 20–59 and ≥ 60 year age groups. SFTS, scrub typhus, and JSF were more common among those aged ≥60 years. More than 60% of cases of Lyme disease affected those aged 20–59 years in both the pre-pandemic and pandemic periods.

The number of new infections for VBDs prevalent in the country stratified by sex was compared between the pre-pandemic (2016–2019) and pandemic periods (2020 only) (Table 4). For relapsing fever and Lyme disease, the incidence in men was approximately twice as high as that in women, and this did not change between the two study periods. For SFTS and JSF, the incidence in men and women was similar before and during the pandemic. For scrub typhus and Japanese encephalitis, the incidence in men was higher than that in women both before and during the pandemic.

## Discussion

This study presented data on the incidence of VBDs under conditions of restriction of human mobility during the COVID-19 pandemic. We found that the onset of malaria and dengue fever showed significant decrease in the first year of the pandemic, and those may be positively correlated with changes in the number of arrivals to the country. On the other hand, no change in the incidence of SFTS, scrub typhus, and JSF was observed during the pre-pandemic and pandemic periods. Their epidemiological curves showed more seasonal patterns rather than correlating with behavioral changes in the domestic population.

A prolonged decline in the incidence of dengue was reported across several dengue-endemic regions at the time when COVID-19-related restrictions were imposed [31]. The decline in dengue cases in 2020 was associated with changes in human movement behaviors, excluding the climatic and immunological factors. Similarly, the incidence of malaria was significantly lower than that predicted during the pandemic period in China [32] and was suspected to be related to the decreased migration of seasonal workers. Since Japan is a non-endemic area for VBDs and surrounded by sea, the disease is usually not spread to the country

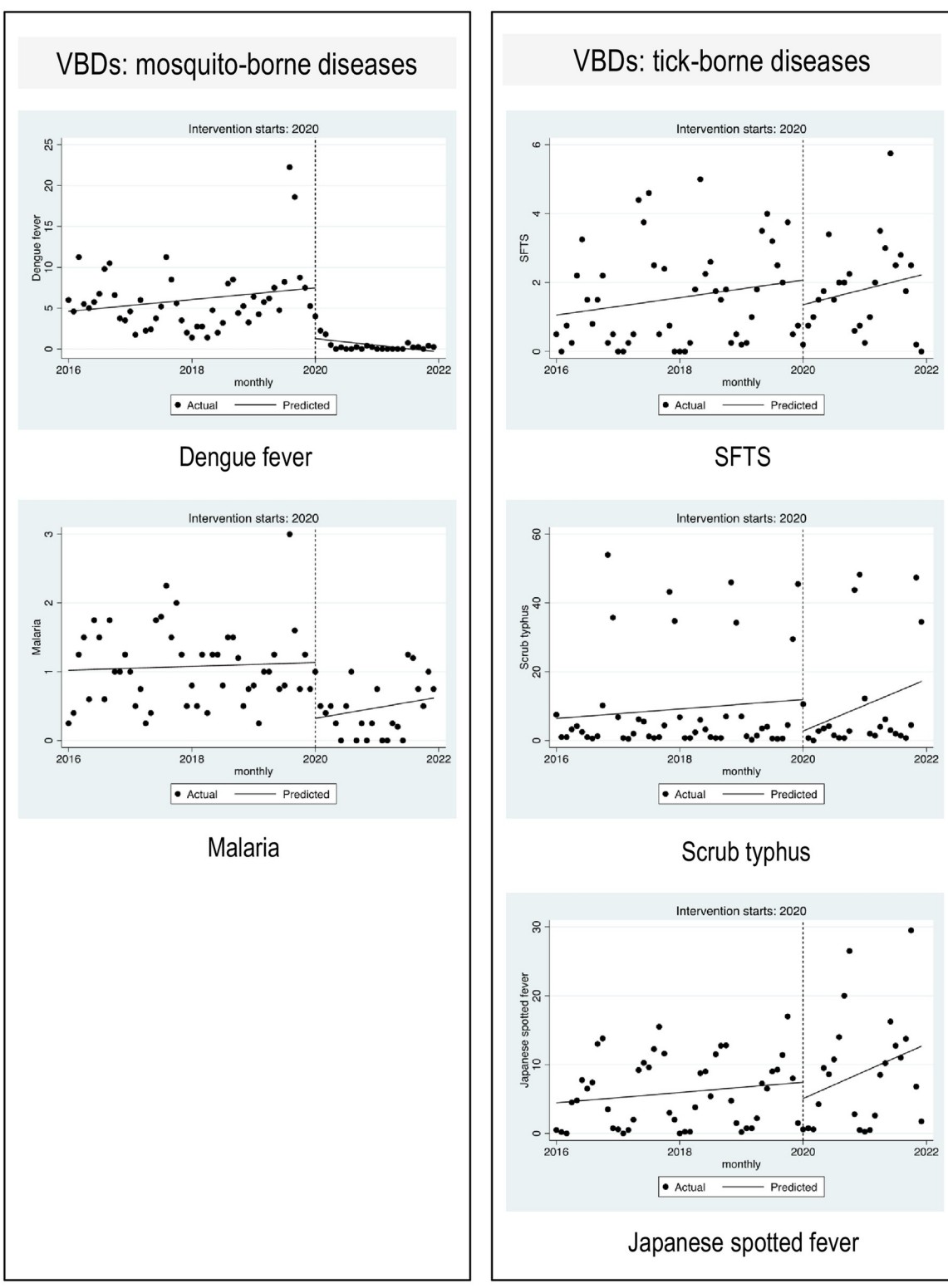

**Fig 3. Interrupted time series analysis conducted for monthly incidences of representative vector-borne diseases between the pre-pandemic and pandemic periods.**

**Table 3. Change in the number of new cases of representative vector-borne diseases by region of infection before and during the COVID-19 pandemic.**

| World region | Zika fever | | Chikungunya fever | | Dengue fever | | Malaria | |
|---|---|---|---|---|---|---|---|---|
| | pre-pandemic | pandemic | pre-pandemic | pandemic | pre-pandemic | pandemic | pre-pandemic | pandemic |
| Asia | 8 (40.0) | 1 (100.0) | 66 (91.7) | 3 (100.0) | 1,091 (87.3) | 43 (81.1) | 32 (14.2) | 6 (12.8) |
| Africa | 0 (0.0) | 0 (0.0) | 1 (1.4) | 0 (0.0) | 14 (1.1) | 1 (1.9) | 169 (75.1) | 32 (68.1) |
| Americas | 9 (45.0) | 0 (0.0) | 1 (1.4) | 0 (0.0) | 27 (2.2) | 2 (3.8) | 2 (0.9) | 0 (0.0) |
| Oceania | 1 (5.0) | 0 (0.0) | 0 (0.0) | 0 (0.0) | 27 (2.2) | 1 (1.9) | 7 (3.1) | 0 (0.0) |
| Europe | 0 (0.0) | 0 (0.0) | 0 (0.0) | 0 (0.0) | 0 (0.0) | 0 (0.0) | 0 (0.0) | 0 (0.0) |
| Other[†] | 2 (10.0) | 0 (0.0) | 4 (5.6) | 0 (0.0) | 90 (7.2) | 6 (11.3) | 15 (6.7) | 9 (19.1) |

| Local region | Relapsing fever | | SFTS | | Scrub typhus | | Tick-borne encephalitis | | JSF | | Japanese encephalitis | | Lime disease | | Epidemic typhus | | Tularemia | |
|---|---|---|---|---|---|---|---|---|---|---|---|---|---|---|---|---|---|---|
| | pre-pandemic | pandemic | pre-pandemic | pandemic | pre-pandemic | pandemic | pre-pandemic | pandemic | pre-pandemic | pandemic | pre-pandemic | pandemic | Pre-pandemic | Pandemic | Pre-pandemic | pandemic | pre-pandemic | pandemic |
| Hokkaido | 23 (82.1) | 25 (96.2) | 0 (0.0) | 0 (0.0) | 0 (0.0) | 0 (0.0) | 4 (100) | 0 (0.0) | 0 (0.0) | 0 (0.0) | 0 (0.0) | 0 (0.0) | 32 (56.1) | 44 (88.0) | 0 (0.0) | 0 (0.0) | 0 (0.0) | 0 (0.0) |
| Eastern Japan | 4 (14.3) | 0 (0.0) | 1 (0.3) | 0 (0.0) | 755 (41.7) | 423 (39.5) | 0 (0.0) | 0 (0.0) | 45 (3.6) | 40 (4.4) | 2 (8.7) | 0 (0.0) | 14 (24.6) | 2 (4.0) | 0 (0.0) | 0 (0.0) | 0 (0.0) | 0 (0.0) |
| Western Japan | 1 (3.6) | 1 (3.8) | 328 (98.7) | 187 (100.0) | 1,057 (58.3) | 649 (60.5) | 0 (0.0) | 0 (0.0) | 1,192 (96.4) | 868 (95.6) | 21 (91.3) | 8 (100.0) | 11 (19.3) | 4 (8.0) | 0 (0.0) | 0 (0.0) | 0 (0.0) | 0 (0.0) |

The numbers in parentheses indicate the percentage of each region within each distribution.

[†]Includes cases with non-definitive suspected source countries/areas or cases who visited multiple countries/areas.

JSF: Japanese spotted fever, SFTS: severe fever with thrombocytopenia syndrome

**Table 4. Percentage of new cases of domestic vector-borne diseases by age group and sex in the pre- and post-COVID-19 pandemic periods.**

| | Period | Age (years) | Relapsing fever | SFTS | Scrub typhus | Tick-borne encephalitis | JSF | Japanese encephalitis | Lyme disease | Epidemic typhus | Tularemia |
|---|---|---|---|---|---|---|---|---|---|---|---|
| Age-group | Pre-pandemic | ≤10 | 0.0 (0) | 0.9 (0.6) | 25.2 (2.8) | 0.0 (0.0) | 21.4 (3.6) | 0 (0.0) | 2.0 (8.5) | 0.0 (0.0) | 0.0 (0.0) |
| | | 20–59 | 0.7 (49.7) | 23.2 (15.5) | 233.3 (26.1) | 2.7 (23.7) | 122.9 (20.5) | 2.7 (23.7) | 14.7 (62.5) | 0.0 (0.0) | 0.0 (0.0) |
| | | ≥60 | 0.7 (50.3) | 125.8 (83.9) | 654.8 (72.3) | 8.5 (76.3) | 455.7 (76.0) | 8.5 (76.3) | 8.9 (37.5) | 0.0 (0.0) | 0.0 (0.0) |
| | Pandemic | ≤10 | 0 (0.0) | 0 (0.0) | 3.6 (1.4) | 0 (0.0) | 2.7 (1.5) | 0 (0.0) | 0 (0.0) | 0.0 (0.0) | 0.0 (0.0) |
| | | 20–59 | 3.56 (45.9) | 4.7 (13.9) | 53.0 (21.3) | 0 (0.0) | 37.1 (20.1) | 0 (0.0) | 8.7 (61.3) | 0.0 (0.0) | 0.0 (0.0) |
| | | ≥60 | 4.2 (54.1) | 29.1 (86.1) | 193.6 (77.7) | 0 (0.0) | 145.6 (79.0) | 2.2 (100.0) | 5.5 (38.7) | 0.0 (0.0) | 0.0 (0.0) |
| Sex | Pre-pandemic | Male | 18.0 (68.2) | 165.0 (52.1) | 1,055.0 (60.0) | 3.0 (76.9) | 573.0 (48.0) | 14.0 (62.2) | 30 (69.3) | 0.0 (0.0) | 0.0 (0.0) |
| | | Female | 8.4 (31.8) | 151.8 (47.9) | 703.4 (40.0) | 0.9 (23.1) | 620.9 (52.0) | 8.5 (37.8) | 13.3 (30.7) | 0.0 (0.0) | 0.0 (0.0) |
| | Pandemic | Male | 9.0 (61.2) | 45.0 (59.8) | 310.0 (59.9) | 0.0 (0.0) | 208.0 (51.0) | 3.0 (61.2) | 18 (67.9) | 0.0 (0.0) | 0.0 (0.0) |
| | | Female | 5.7 (38.8) | 30.3 (40.2) | 207.4 (40.1) | 0.0 (0.0) | 199.8 (49.0) | 1.9 (38.8) | 8.5 (32.1) | 0.0 (0.0) | 0.0 (0.0) |

Figures in parentheses indicate the percentage of each sex for each disease. [†]Patient characteristics for the year 2021 were not available and were thus excluded from the analysis. The figures are corrected according to the proportion of each age group / each gender in the census population. JSF: Japanese spotted fever, SFTS: severe fever with thrombocytopenia syndrome

except by entry from abroad, the association between transborder restrictions on human mobility and the marked reduction in the incidences of these VBDs during the pandemic period may have been more strongly evident.

Among the imported cases of VBDs reviewed in this study, several diseases, such as Zika and Chikungunya, had very low incidence rates during the COVID-19 pandemic period, which may not be directly linked to a decrease in the number of people entering the country [33, 34]. More than 80% of foreign visitors during the pre-pandemic period were Asians, indicating a high risk of importing malaria, dengue, Zika, and Chikungunya fevers, which are endemic to Asia, into Japan [35–38]. During the pre-pandemic period, such infectious diseases were brought in from Asian countries. However, the number of immigrants from Asia to Japan during the pandemic period declined by -92% compared to the pre-pandemic period. Thus, in the case of dengue fever or Chikungunya, the decline in the number of Asian immigrants may be associated with the incidence rates of these diseases during the pandemic in Japan. For malaria, 75.1% of cases were brought in from African countries during the pre-pandemic period. Although there was no significant decrease in the percentage of malarial infection among African visitors during the pandemic period, there was a significant decrease in the number of cases per year, from 42.25 (pre-pandemic) to 16.0 (pandemic). Thus, the decline in this number among African visitors may have a significant impact on the incidence of malaria in Japan. Therefore, fluctuations in international travel and immigration from endemic regions have the potential to change the domestic incidence [39]. Additionally, no VBDs cases were brought in from North America or Europe throughout the study period, although the percentage of visitors from Europe increased, surpassing visitors from Asian countries in July 2021. It is not clear from this study whether the absence of infected travelers

from Europe was due to low incidences of dengue fever, malaria and other VBDs in Europe or because the proportion of travelers was generally small.

The SFTS, scrub typhus, and JSF showed no significant changes in terms of their incidences before and after the pandemic following ITSA. Moreover, their epidemic curves showed yearly cyclic change during the pandemic period and did not seem to correlate with human mobility, at least not with mobility to the park. These VBDs are tick-borne diseases. This would suggest that tick-borne diseases were not affected by changes in local mobility. However, a Taiwanese study using national data showed a significant decrease (52% decrease) in the incidence of scrub typhus disease in rural areas and remote islands in 2021, suggesting that a decrease in travelers to such regions may be the cause [40].

A study of SFTS cases in Japan reported that the most frequent activity at the time of infection was farming (65%) [41]. A national study of scrub typhus in Japan showed that the activity engaged in at the time of infection was acquired included farming (31.6%) and forestry work (14.2%) [42]. Another study of JSF reported that farming (35.7%) was the most frequent activity [43]. Thus, SFTS, scrub typhus, and JSF have little to do with spatial activities such as traveling but rather with daily activities in Japan. Since these usual activities were unlikely to have been affected by the COVID-19 pandemic, it is consistent that the epidemic curves of these VBDs were similar to their pre-pandemic curves. This claim may be supported by the fact that if tourists and others short-term visitors become infected with the tick-borne pathogen in the endemic area and, after an incubation period, develop the disease in the urban area, this will lead to an increase in the incidence in urban area. However, infection with VBDs occurred within the reporting prefecture for majority of the cases [14].

In the East Asia, the elderly group aged 60 years or older was the most common age group in patients with SFTS, scrub typhus, and JSF [41–44]. Similarly, in our study, most patients with these diseases were in the elderly group throughout the two study periods. The average age of Japanese farmers was 67.8 years in 2020, and 69.6% of farmers were aged ≥65 years in 2020, making this age group the core of independent farm management in Japan, which was similar to situation in Korea [45] and was significantly elderly compared to Western countries [46]. We found also no significant change in the sex distribution for each VBD predominant in the country between the pre-pandemic and pandemic periods. This may suggest that the population subject to infection risk behaviors, such as agricultural and forestry workers, did not change significantly between these periods.

This study showed no significant changes in the suspected region of infection. SFTS and JSF were dominant in Western Japan, whereas scrub typhus was dominant in Eastern and Western Japan. For scrub typhus and JSF, the pathogens were transmitted between vectors via gametes rather than through ticks sucking blood from an infected person or animal [47]. This suggested that the spread of the infection to an area is less likely to be affected by human mobility, and that the suspected region of infection corresponds to the vector's habitat area. Therefore, an unchanged suspected region of infection may suggest that the vector's habitat area did not change significantly during the COVID-19 pandemic.

As indicated in this study, Lyme disease was more prevalent in Hokkaido, with some cases occurring in areas outside Hokkaido [48]. These vectors can be found on the plains in Hokkaido, but outside Hokkaido, they are mainly found in alpine areas (900–1700 m<) [49]. If Lyme disease is more commonly encountered during mountain climbing outside Hokkaido, the decline in climbers during the pandemic may have affected the decline in the incidence of Lyme disease outside Hokkaido [50].

In this study, the incidence of scrub typhus and JSF were shown an increasing trend in the pandemic period. Fielding et al. reported an increase in migration to rural areas away from metropolitan areas in Japan during the pandemic, using inter-prefectural population movement

data [51]. This was thought to reflect the growing interest in moving to and working in rural areas to avoid living in densely populated regions during the COVID-19 pandemic [51]. Moreover, we cannot deny that local mobility restrictions for the COVID-19 pandemic could lead to increases in VBDs by disrupting medical services and vector control campaigns [52].

This study showed the decline in access to healthcare during the COVID-19 pandemic period. Certainly, internal medicine and pediatrics showed a large refrain rate; however, the reduction in dermatology visits was less in this study. In addition, there may be cases of refusal or postponement of consultations by medical personnel. Therefore, it is difficult to calculate the true number of cases of refusal to visit a healthcare center and to eliminate the impact of such refusal. However, the major mosquito-borne infectious diseases, dengue fever, Chikungunya fever, and Zika fever, if manifested, are characterized by fever and a generalized rash [53]. These infections are difficult to distinguish based on clinical symptoms. For tick-borne diseases, scrub typhus (n = 4,185) and JSF (n = 1,765), fever was observed in 95% and 99% of cases and rash in 86% and 94% of cases, respectively [54]. It is conceivable that the medical consultation rates for such emerging symptoms are not low. In addition, the majority of patients with tick-borne diseases were elderly individuals. Therefore, it is likely that the number of visits to the pediatric department was quite low, even during the non-pandemic period. In endemic areas, a high proportion of visits to dermatologists were considered to be made if there was a history of tick bites. Therefore, we concluded that the effect of avoiding visiting a healthcare center was relatively small.

As discussed, there were a variety of correlated factors for the incidence of VBDs [Fig 4]. Human mobility appeared to have more impact on mosquito-borne infectious diseases but little impact on tick-borne diseases. Instead, geographical or seasonal factors had a greater impact for tick-vector diseases. Host factors such as age and sex differences were found to affect tick-borne diseases, but biological, social and behavioral factors must also be taken into account to determine the true impact. Mosquito-borne diseases, such as malaria, are reportedly modified by behavioral differences between men and women [55]. Commodity distribution was considered to have a small impact on both diseases. However, mosquitoes can be spread around other areas of the world by ship transport [56]. Other correlating factors were not considered in this study, although, it should be noted that the influence of each correlated factor on the incidence of mosquito- and tick-borne diseases varies between countries and regions. For example, in countries with distinct wet and dry seasons, climatic factors may have a strong influence on mosquito-borne disease outbreaks beyond seasonal effects [57]. Moreover, animals with ticks move into residential areas, expanding the distribution area of various ticks [58].

## Study limitations

VBDs have a distinct seasonality, as shown in this study. It was, therefore, important to eliminate the influence of seasonality. Japan is made up of islands of various sizes, and its length, from north to south, is over 2,000 km [59]. The climate is not uniform, ranging from subtropical in the south to subarctic in the north [59]. The presence of the Blakiston's and Itoigawa-Shizuoka Tectonic Lines, which divide the insect-living areas, is also evidence of the diversity of climate. For example, regarding scrub typhus, the epidemics occur throughout the country, but the peak of the epidemic varies by region because of regional differences in its vector species [54]. Therefore, it was difficult to eliminate seasonality or climatic correlating factors in the present study using national data. In the future, we intend to examine the influence of climate in a regionally restricted study.

The onset of VBDs was subject to the level of population immunity gained through previous infections. However, this study was a national survey using a national data base and

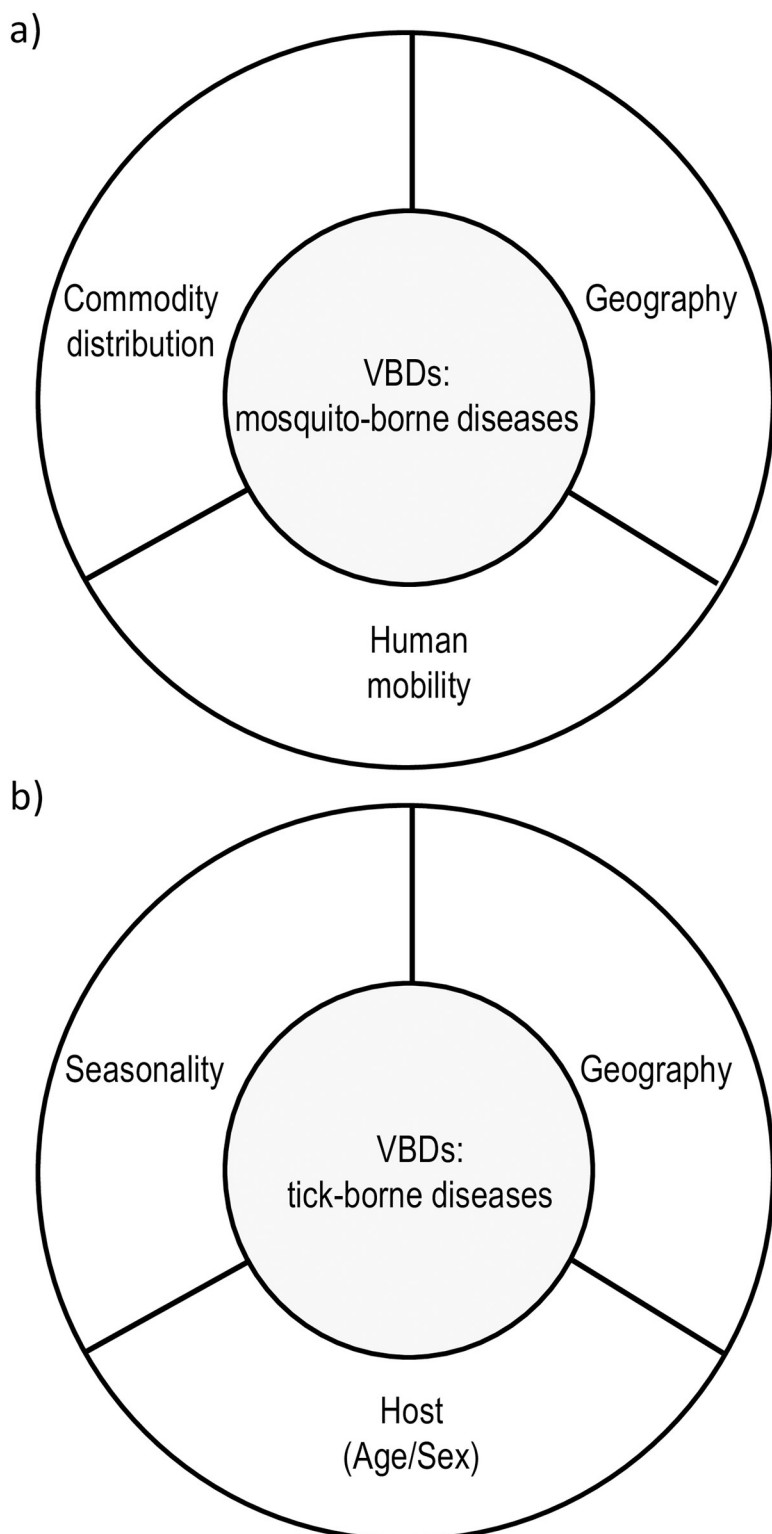

**Fig 4. The hypothesized correlating factors influencing the incidence of vector-borne diseases.** VBDs: Vector-Borne diseases.

antibody levels in the population were not available; therefore, the impact of this could not be excluded. However, there was a study that examined antibody levels in local populations in an endemic area for SFTS [60]. In this study, one out of 694 individuals tested positive for antibody titers. This means that even in endemic areas, the local population is unlikely to be immune to SFTS. The rate of rickettsia-bearing toxic ticks in scrub typhus (*Leptotrombidium* sp.) was estimated to be 0.1–3% [53]. The percentage of antibodies to scrub typhus was, therefore, considered to be quite low. However, this is an issue to be considered in the future.

## Conclusions

The COVID-19 pandemic resulted in different incidences of mosquito-borne and tick-borne diseases. This incidence of mosquito-borne diseases showed a significant reduction with the advent of the pandemic; however, transborder mobility restrictions may influence the incidence. In contrast, the incidence of tick-borne diseases was not significantly different before and after the pandemic but rather remained seasonal. This suggested that tick-borne diseases were unlikely to be affected by domestic movement restrictions.

## Supporting information

**S1 Fig. Indices of tertiary industry activity (base year is 2015 and is set at 100) and the monthly number of newly confirmed COVID-19 cases.** a) Indices of travel agency activity. b) Indices of the passenger transportation industry activity. The declaration of a state of emergency was implemented from April 7 to May 25, 2020; from January 8 to March 21, 2021; from April 25 to June 20, 2021; and from July 12 to September 30, 2021. The quasi-emergency measure was implemented from April 5 to September 30, 2021, and from January 9 to March 1, 2022.
(PPTX)

**S2 Fig. Number of receipts by department for medical clinics compared to the same month in 2019.** Figures indicate the number of outpatients who visited the clinic by month. The declaration of state of emergency was in effect from April 7 to May 25, 2020; from January 8 to March 21, 2021; from April 25 to June 20, 2021; and from July 12 to September 30, 2021. In addition, quasi-emergency measures were in effect from April 5 to September 30, 2021.
(PPTX)

**S3 Fig. Changes in the movement of people and the volume of goods transported from overseas before and after the COVID-19 pandemic.** (a) Changes in the number of foreign nationals entering Japan between January 2016 and December 2021. The filled rectangle represents the period of the Olympic Games in Tokyo (from July 23 to August 8, 2021). The unfilled rectangle represents the period of the Paralympic Games in Tokyo (from August 24 to September 5, 2021). (b) Changes in the number of Japanese returnees to Japan from January 2016 to December 2021. (c) Changes in the actual amount of loaded container cargo volume handled within foreign trade from 2016 to 2021.
(PPTX)

**S4 Fig. Current epidemic curves for vector-borne diseases since 2016 in Japan.** SFTS: severe fever with thrombocytopenia syndrome.
(PPTX)

**S1 Table. Tertiary industry activity index (monthly) by industry (2015 = 100.0%) and the number of newly confirmed cases of COVID-19 per 100,000 population.** *Figures indicate

the numbers of newly confirmed cases of COVID-19 per 100,000 population.
(XLSX)

**S2 Table. Number of people entering the country (foreign arrival and returnees).**
(XLSX)

**S3 Table. Number of containerized cargo handled in external trade.**
(XLSX)

**S4 Table. Domestic passenger and freight transport distances by mode of transport.** *Indicates passenger volume/passenger-km on the Bullet Train, excluding regular ticketed passengers. †Indicates the number of passengers transported on scheduled domestic flights. mil: million, n.a.: Not yet officially announced.
(XLSX)

**S5 Table. Changes in the population staying in parks and residential areas.** Figures indicate how the number of visitors to parks and residential areas has changed compared to baseline days (the averages from the 5-week period from January 3 to February 6, 2020).
(XLSX)

**S6 Table. Percentage reduction in access to medical care during the COVID-19 pandemic.** Figures are calculated based on clinic outpatient visits vs. the same month in 2019.
(XLSX)

**S7 Table. Number of newly infected cases per month for vector-borne diseases under the notifiable surveillance system.**
(XLSX)

**S8 Table. Number of new cases of vector-borne infectious diseases by year and estimated transmission area.** †Includes unknown cases with regard to suspected infected countries/areas. SFTS: severe fever with thrombocytopenia syndrome.
(XLSX)

**S9 Table. Number of new cases of vector-borne diseases by estimated regions of acquired infection.** †Including visitors from more than one country and those from unknown infected areas. Italics include values for returnees. A-NZ: Australia & New Zealand, C: Central, E: Eastern, M: Middle, N: Northern, S: Southern, S-E: South-Eastern, W: Western, Mela: Melanesia, Micro: Micronesia, Poly: Polynesia. SFTS: severe fever with thrombocytopenia syndrome.
(XLSX)

**S10 Table. Number of new cases of endemic vector-borne diseases by sex and age.** Figures are corrected according to sex ratio/age group ratios based on the national census. The number of females was calculated by multiplying the number of females by the ratio of males to females. For example, the number of females in 2016 was calculated by multiplying the gender ratio for that year by 0.948. SFTS: severe fever with thrombocytopenia syndrome.
(XLSX)

## Acknowledgments

We would like to thank Editage (www.editage.com) for English language editing.

## Author Contributions

**Data curation:** Akira Shinzato, Takeshi Kinjo.

**Formal analysis:** Hiroyoshi Iwata.

**Supervision:** Masao Tateyama, Kazuko Yamamoto, Jiro Fujita.

**Writing – original draft:** Kenji Hibiya.

**Writing – review & editing:** Kazuko Yamamoto, Jiro Fujita.

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
