## [Decision Letter · Decision Letter 0]

21 Nov 2022

PONE-D-22-18599Effect of human mobility restriction on vector-borne diseases during the COVID-19 pandemic in Japan: A descriptive epidemiological study using a national database (2016 to 2021)PLOS ONE

Dear Dr. Hibiya,

Thank you for submitting your manuscript to PLOS ONE. After careful consideration, we feel that it has merit but does not fully meet PLOS ONE’s publication criteria as it currently stands. Therefore, we invite you to submit a revised version of the manuscript that addresses the points raised during the review process.

ACADEMIC EDITOR: Please note that we have needed three reviewers to decide on your manuscript, so we ask that you carefully consider the suggestions. Please note that citation suggestions will not influence the final decision on your manuscript.

We look forward to receiving your revised manuscript.

Kind regards,

Kovy Arteaga-Livias

Academic Editor

PLOS ONE

https://journals.plos.org/plosone/s/fileid=ba62/PLOSOne_formatting_sample_title_authors_affiliations.pdf.

2. We note that [Figure 1] in your submission contain [map/satellite] images which may be copyrighted. All PLOS content is published under the Creative Commons Attribution License (CC BY 4.0), which means that the manuscript, images, and Supporting Information files will be freely available online, and any third party is permitted to access, download, copy, distribute, and use these materials in any way, even commercially, with proper attribution. For these reasons, we cannot publish previously copyrighted maps or satellite images created using proprietary data, such as Google software (Google Maps, Street View, and Earth). For more information, see our copyright guidelines: http://journals.plos.org/plosone/s/licenses-and-copyright.

Natural Earth (public domain): http://www.naturalearthdata.com/.

Reviewers' comments:

Reviewer's Responses to Questions

**Comments to the Author**

1. Is the manuscript technically sound, and do the data support the conclusions?

Reviewer #1: Partly

Reviewer #2: Partly

Reviewer #3: Partly

2. Has the statistical analysis been performed appropriately and rigorously? 

Reviewer #1: No

Reviewer #2: No

Reviewer #3: I Don't Know

3. Have the authors made all data underlying the findings in their manuscript fully available?

Reviewer #1: Yes

Reviewer #2: Yes

Reviewer #3: Yes

4. Is the manuscript presented in an intelligible fashion and written in standard English?

Reviewer #1: Yes

Reviewer #2: Yes

Reviewer #3: No

5. Review Comments to the Author

Reviewer #1: This manuscript uses time series data for vector-borne diseases in Japan to ask whether Covid-related mobility changes led to significant changes in vector-borne disease cases. The authors find a significant decrease in imported vector-borne diseases, particularly malaria. Most endemic diseases exhibited no significant change, except for SFTS and scrub typhus, which significantly increased.

Estimating how behavioral changes connected to Covid-19 affected transmission of other diseases is important for planning for future pandemic and understanding drivers of vector-borne disease transmission. The authors use extensive data collected between 2016 and 2021 and examine several potentially relevant covariates. However, I do not think the current analyses and conclusions are appropriate for publication. In particular, I have questions about the appropriateness and robustness of the statistical methods used.

Major:

It is not clear from the methods, but it appears that the interrupted time series analysis was conducted by aggregating points across a year, meaning that there were four points in the pre-pandemic period and two points in the post-pandemic period, and extremely small sample size. The interrupted time series analysis also assumes a linear trend over time, but no information is provided to support this assumption. There is also not sufficient information provided about the coefficients returned from the ITSA and their significance to evaluate the results of this analysis.

Additionally, the assumptions underlying this analysis require further justification. For example, there is often some lag in vector-borne disease transmission, so it’s unclear that the effects of Covid-related behavioral changes would be observed immediately. Although the analysis assumes stationarity in the pandemic period, the types and intensity of Covid-related behavioral changes may also have changed over the course of the 2020-2021 time period. It’s also not clear why the pre-pandemic period is defined as starting in 2016 and the pandemic period ends in 2021 for this analysis. Is this due to data access or a methodological decision? Is it accurate to assume stationarity across each of these periods?

Covid may have also disrupted surveillance efforts and decreased access to healthcare, leading to a change in reporting rates. This effect could significantly influence your results, especially the static comparisons between pre and post-pandemic disease burden. This should be investigated directly (perhaps by comparing metrics that could serve as proxies for reporting rate, like test positivity and non-Covid medical appointments in the pre and post-pandemic periods) as it could significantly impact the study’s findings.

The Olympic Games are mentioned several times, but it’s unclear what the hypothesized effect is and what VBD and Covid precautions (if any) were taken while the games happened. It may even be worth considering whether the Olympic Games themselves constitute another event that may have separately altered VBD transmission. Line 356 states that there was no “no significant increase” during the Olympic Games, but doesn’t include an analysis to support this claim.

In general, the manuscript considers several diseases and covariates, which can make the paper difficult to follow. Better organization throughout and a schematic figure that illustrates the key questions/proposed mechanisms could help to guide the reader.

Below, I mentioned a few cases where analyses and their corresponding hypotheses are not in the introduction, as well as strong causal claims that are made without sufficient evidence. Such claims should be supported with either additional analyses or references to other studies.

Minor:

line 19: “common” should be “domestic” or “endemic” (and be consistent throughout)

line 22: clarify that the analysis was conducted using data for Japan at the regional level

line 33: change semicolon to period

line 44: were there restrictions on domestic travel? Please also include discussion of when restrictions were implemented and how long they were in place

line 45: remove “unnecessary”

line ~ 49: You should also mention that Covid restrictions could lead to increases in vector-borne diseases by disrupting medical services and vector control campaigns.

Line 59: mention Covid impacts on trade here

Line 63: explain why you would expect a difference between short and long-term effects

Line 65: “since the COVID-19 pandemic” – arguably, the pandemic is ongoing, so it may make more sense to say “since the COVID-19 pandemic began”

In the introduction, you should also discuss domestic mobility and where humans tend to come into contact with the different types of vectors to develop hypotheses for how diseases may have changed during the pandemic based on mobility changes.

In the introduction, you should introduce the ideas behind the estimated infected areas analysis, gender, and age-specific incidence analysis, why you think the distribution may have changed under Covid, and the hypothesized direction of that change.

Throughout the manuscript, you use the term “mobility restriction” generally, but it would be useful to clarify when you’re discussing bans on international travel vs reduction in local travel, as the expected outcomes differ.

Line 122 – what percent of people riding bullet trains are commuter (are you excluding a significant proportion of train riders from your analysis?)

Line 131-136 – is there precedent for using passenger-kilometer and ton-kilometer in this manner as metrics for passenger/cargo transport? Please cite other studies that use them.

Line 150 – “estimated regions of infection” please explain in greater detail what this means and how it was determined. Is it based on individual follow-up for each case?

Line 159 – you should also mention in this paragraph that you have data on whether particular cases of each disease were transmitted locally or imported and explain how that information was initially assessed.

Line 174 – clarify that you are comparing annual burden in either period (otherwise it’s confusing to compare a 4 vs 2 year period)

Line 177-179 – is this part of the analysis described in the previous paragraph? If so, make that clear

Line 205 – clarify it was “on the rise” prior to the pandemic’s beginning

Line 205-235 – a lot of the information given here feels extraneous, which makes it difficult to keep track of the general trends that you’re describing in this section

Fig 3 – add titles and y-axis labels to each of the plots to make it easier to understand what each panel is plotting

All tables – note any significant differences in bold

Line 250 – “an overall decline” -> “fewer cases”

Line 256 – “statistically” -> “statistical”

Line 256-258 – “malaria showed significantly…Japanese spotted fever” need more detail here. What were the estimated slopes and intercepts? It also seems like you’re missing discussion of dengue and Japanese spotted fever.

Lines 282 and throughout – try to avoid using “significant” if you’re not referring to the result of a statistical analysis

Fig 4 – It’s inconsistent to have an average for 2016-2019 and then the yearly values for 2020 and 2021. Can you show just individual years or just averages for all of them (although that may get a bit crowded)? I also think you can move some of the panels to the supplement and just focus on a few of the disease that showed the most interesting/significant trends with this figure.

Table 3 - for pre-pandemic SFTS in West Japan, it looks like the percentage is wrong (should be 99.7)

Line 320 – “was suppressed” I don’t think you actually showed this statistically. If so, please reference the relevant analysis. I would also change the order of this to lead with your most important findings.

Line 322 – “Maralia” -> “malaria”

Line 339 – “was only 0.87% of that in 2019” this is quite low, so make sure the number is correct and also cite your source/the table you’re referencing for this.

Line 347-355 This paragraph is confusing. What do you mean by saying trade is “currently saturated and unlikely to increase/decrease?” How does it compare to the pre-pandemic baseline or the time period observed in the study? “A rapid epidemic would have occurred…” is an extremely strong claim, and there are no analyses or sources here to support it.

Line 357 – “We attribute this” you don’t have sufficient evidence to make this causal claim

Line 363 – “However, the number of patients…” -> “However, a change in the number of patients”

Line 367 – “travel for tourism purposes” explain more why you think this would have changed the distribution of travel and provide sources.

Line 378-383 – too much detail provided here about someone else’s study

Line 398 – “ these may be plausible” please try to find additional sources to support this hypothesis. Have other people found similar trends?

Line 404 – “ humans are not an infectious source” – what does this mean? Humans can’t transmit back to the vector?

Line 406 – “suspected region of infection” relate this back to a specific table and cite it

Line 423 – “various studies” -> cite the studies you’re referencing

Line 443 – “new residents in rural areas” - do regional patterns bear this out? What other data and tables from the study can you cite to support this?

Line 461 – “new infectious diseases such as COVID-19” this sentence is confusing because Covid isn't a vector-borne disease

Reviewer #2: In this study, the authors examined the impact of mobility restriction on vector-borne diseases during the COVID-19 pandemic in Japan through a descriptive epidemiological study. They collected both mobility data and several typical vector-borne diseases data. They performed a simple interrupted time series analysis. I have a few comments:

1. In line 41, “Wuhan (Zhejiang Province, People's Republic of China)”. This might be corrected as “Wuhan (Hubei Province, People's Republic of China)”.

2. Vector-borne diseases have a strong seasonality and subject to the level of population immunity gained through previous infections [ref 24]. The authors may wish to account for these confounding factors. Otherwise, it is difficult to understand if the observed pattern in incidence is due to restrictions or other factors.

3. In line 174, “estimated number of infections”. Could you explain how the number of true infections was estimated in this study?

Reviewer #3: This manuscript, “Effect of human mobility restriction on vector-borne diseases during the COVID-19 pandemic in Japan”, provides a nice summary of how the incidence of imported and domestic vector-borne disease cases have shifted during the COVID-19 pandemic period in pandemic compared to the pre-pandemic period. The analysis includes all vector-borne diseases that are nationally notifiable, which allows the authors to look at broader trends that might be missed in an analysis of a single pathogen. The manuscript is labeled as a descriptive epidemiological study, however, I believe that the statistical analyses that are conducted need improvement. In addition, some of the results need to be more clearly presented as in some sections of the Results and Discussion it is stated that several of the domestic diseases increased during the pandemic, but then in another section it will say that the same diseases did not experience a statistically significant change in incidence. If this is due to differences in the results of the different statistical tests (t-test vs. ITSA) then which test is being referred to needs to be clearly stated.

Major comments

1. No values are reported for any of the statistical analyses (e.g., the t-score and p-values for the t-tests, or p-value and coefficient estimates for ITSA). It is also not clear which analysis (the t-test or ITSA) is being referenced in some of the Results (e.g., on lines 241-244), including the appropriate statistical values will make it clearer which test is being mentioned.

2. For the Interrupted Time Series Analyses (ITSA), it is not clear whether the results presented are for significant changes in the level or trend pre- and post-pandemic. Both the Methods and Results sections need to specify which effect is being reported. In addition, because there are only four pre-pandemic data points, and only 2 pandemic data points, testing for changes in trends would be statistically inappropriate. There are also doesn’t appear to be a clear scientific reason why you would want to test for a change in the trend between the two time periods. The Methods section should also mention that the ITSA was conducted using annual incidence (as opposed to the t-test which mentions using weekly incidence data).

3. It is also not clear whether the ITSAs were carried out separately for each disease or if multiple diseases were modeled as part of a multiple-group comparison. I assume that each disease was modeled separately, but line 185 refers to “multiple-group ITSA”, which would imply that the authors were also comparing the response of different diseases to the pandemic “intervention”. This section needs clarification one way or the other, and if a multiple-group analysis was conducted, the reasons for such a choice should be articulated.

4. In multiple sections (Abstract, Results, and Discussion) there are contrasting statements regarding whether certain diseases increased or stayed the same during the pandemic compared to the pre-pandemic period. For example, line 31-33 of the abstract says that SFTS, scrub typhus, and Japanese spotted fever remained similar and then that SFTS and scrub typhus increased significantly in the very same sentence.

Minor comments

This study might benefit from the inclusion of cell-phoned derived mobility data. For example, Google has made data available on visits to multiple types of locations (https://www.google.com/covid19/mobility/). This could be an additional metric to show how movement rates changed in 2020 and 2021. In particular, one metric included by Google is visits to “parks and other natural areas”, which is relevant for several of the vector-borne diseases as mentioned in the Discussion section.

Line 34 – “pan-endemic”: I assume this should be “pandemic”?

Line 154-55: Japanese encephalitis is missing from this list of domestic vector-borne diseases included in the study.

Line 178 – reference to Table S4 is to the incorrect table

Table 1 is never referenced in the body of the text.

The global incidence of several of the considered diseases, particularly Zika and chikungunya, was not stable from 2016-2019 before the COVID-19 pandemic. Zika incidence decreased from 2016-2019 as the epidemic in Latin America subsided. There should be some mention in the Discussion section that global trends might also influence incidence rates for imported diseases.

Lines 257-58 – This sentence is incomplete.

Lines 268-279 – In the “Estimated infected areas of imported infectious diseases” subsection of the Results, the word “in” needs to be replaced with “from” in multiple sentences. As in “cases were reported in Asia only” should be changed to “cases were reported from Asia only”.

Lines 275-277 – 81.1% of cases from Asia is not significantly more common than 87.5% of cases.

Table 3 – The first part of “Relapsing fever” is missing

Line 322 – Malaria is spelled wrong

Line 372 – The start of the “Domestic infectious diseases” subsection of the Discussion discusses how SFTS, scrub typhus, and spotted fever showed no apparent decrease during the pandemic period. It is only several paragraphs later that it is mentioned that these 3 diseases, plus Lyme and relapsing fever, were actually higher during the pandemic period. It makes more sense to mention the increase first rather than simply say that they didn’t decrease.

Line 449 – Should ‘exclude’ be ‘include’?

6. PLOS authors have the option to publish the peer review history of their article (what does this mean?). If published, this will include your full peer review and any attached files.

Reviewer #1: No

Reviewer #2: No

Reviewer #3: No

---

## [Author Response · Author response to Decision Letter 0]

7 Feb 2023

Reviewer 1: 

This manuscript uses time series data for vector-borne diseases in Japan to ask whether Covid-related mobility changes led to significant changes in vector-borne disease cases. The authors find a significant decrease in imported vector-borne diseases, particularly malaria. Most endemic diseases exhibited no significant change, except for SFTS and scrub typhus, which significantly increased.

Estimating how behavioral changes connected to Covid-19 affected transmission of other diseases is important for planning for future pandemic and understanding drivers of vector-borne disease transmission. The authors use extensive data collected between 2016 and 2021 and examine several potentially relevant covariates. However, I do not think the current analyses and conclusions are appropriate for publication. In particular, I have questions about the appropriateness and robustness of the statistical methods used.

Thank you for reviewing our paper and making insightful comments. We have revised the manuscript according to your suggestions. Please find our point-by-point responses below.

Major comments:

C1. It is not clear from the methods, but it appears that the interrupted time series analysis was conducted by aggregating points across a year, meaning that there were four points in the pre-pandemic period and two points in the post-pandemic period, and extremely small sample size. The interrupted time series analysis also assumes a linear trend over time, but no information is provided to support this assumption. There is also not sufficient information provided about the coefficients returned from the ITSA and their significance to evaluate the results of this analysis.

(Response 1) Thank you for your comment. In the preliminary ITSA analysis, we used weekly points, but because there were weeks with many zeros, we used yearly data. However, the reviewer's point is valid, and we have changed it to monthly data. Therefore, S1 Fig has been revised (changed to Fig. 3).We apologize for the lack of information on the linearity assumption and have included it in the Materials and Methods section as follow:

Material and Methods

Statistical analysis (Line 199-210)

Interrupted time series analysis (ITSA) was conducted using monthly data to determine significant changes for the incidences of representative diseases during the pre- and post-COVID-19 pandemic following the method described by Linden [30]. ITSA is a method of evaluating the impact of an intervention on an outcome (the causal effect) in terms of the level of change (the change in intercept) or the trend in change (the change in slope) within a time series [30]. The analysis was focused on diseases whose monthly incidence could be statistically analyzed (an average monthly incidence of >1.5 for domestic endemic diseases) or whose international was high, such as dengue fever, malaria, SFTS, scrub typhus, and Japanese spotted fever (JSF). First, we examined whether there was a linear underlying time trend of the incidences of vector-borne diseases of pre-pandemic and incidence of vector-borne diseases during the pandemic using the Cochran–Armitage test. Ordinary least-squares regression models designed to adjust for autocorrelation were employed for the single-group ITSA, which included data on the pre-pandemic and pandemic periods.

Results (Line 269-281)

We conducted ITSA as statistical analysis to examine significant differences between the pre-pandemic and pandemic periods (Table 2, Fig 3). Before the COVID-19 pandemic began, dengue fever/malaria onset per month did not significantly increase or decrease every month. In the first year of the pandemic (2020), there appeared to be a significant decrease in dengue fever (p=0.003) and malaria (p=0.002) onset per month, followed by a non-significant increase in the annual trend of dengue fever and malaria onset per month. No significant differences were observed for SFTS (p=0.265), scrub typhus (p=0.207), and JSF (p=0.514) during the pre-pandemic and pandemic periods. However, there was a change in slope from 0.021 to 0.016, for SFTS; from 0.115 to 0.511, for scrub typhus; and from 0.062 to 0.268 for JSF.

Table 2. Interrupted time-series analysis of the incidences of vector-borne diseases for the pre- and post-pandemic periods

CI: confidence interval, JSF: Japanese spotted fever, SFTS: severe fever with thrombocytopenia syndrome.

C2. Additionally, the assumptions underlying this analysis require further justification. For example, there is often some lag in vector-borne disease transmission, so it’s unclear that the effects of Covid-related behavioral changes would be observed immediately. Although the analysis assumes stationarity in the pandemic period, the types and intensity of Covid-related behavioral changes may also have changed over the course of the 2020-2021 time period. It’s also not clear why the pre-pandemic period is defined as starting in 2016 and the pandemic period ends in 2021 for this analysis. Is this due to data access or a methodological decision? Is it accurate to assume stationarity across each of these periods?

(Response 2) First, we considered that the lag in transmission of each vector-borne disease is absorbed to some extent in ITSA by taking the observation period beyond that. Second, it was difficult to demonstrate steady-state during a pandemic period without strong restraints on human behavior, such as a lockdown. However, in Taiwan and Japan, moderate interventions were introduced in response to domestic outbreaks of COVID-19. The period of Japanese’s measure is shown in S1 Fig below. The figure shows the activity of the travel agency business and the passenger transport business in Japan. It shows that from 2020 to 2021, people's behavior appeared to be somewhat restrained. However, since the activity of each passenger transportation business has been increasing since the spring of 2022, we believe that people's mobility has increased. Therefore, we assumed some degree of steady state was ensured from 2020 to 2021. Further, we have modified the title of this study to indicate that it is under moderate behavioral restraint. Moreover, we set the study period to begin in 2016 because the number of domestic cases of Chikungunya fever was available from 2016. The pandemic period was set until 2021 because various intervening factors were considered from 2022 owing to the relaxation of behavioral restrictions, the holding of events, and the reopening of international flights and on.

　 　　 

S1 Fig. Indices of tertiary industry activity (base year is 2015 and is set at 100).

a) Indices of travel agency activity. b) Indices of the passenger transportation industry activity. 

Title page (Line 1–3)

Effect of non-coercive human mobility restrictions on vector-borne diseases during the COVID-19 pandemic in Japan: A descriptive epidemiological study using a national database (2016‒2021)

 Introduction (Line 50)

Forced restriction of human mobility seemed to have reduced the incidence of VBDs [7]. 

Material and Methods

Study design and study periods (Line 82-99)

…..We set the study period to begin in 2016 because the number of domestic Chikungunya fever cases was available from 2016 in Japan. The pandemic period was set until 2021 because various confounding factors were considered from 2022 owing to the relaxation of behavioral restrictions, holding of events, and reopening of international flights. We examined the activity of the tourism industry and passenger transport services from January 2019 to September 2022 during the COVID-19 pandemic in Japan (S1 Fig.). Data were obtained from monthly reports of Indices of Tertiary Industry Activity released by the Ministry of Economy, Trade and Industry [15] (S1 Table). It showed that from 2020 to 2021, people's mobility appeared to be somewhat restrained. However, since the activities of passenger transport services had increased since the spring of 2022, we assumed that people's activity increased. Therefore, we assumed that some level of stability was maintained from 2020 to 2021. In addition, the declaration of a state of emergency was the highest behavioral restriction to prevent the spread of COVID-19 during the pandemic period in Japan (S1 Fig.). The national and local governments requested residents to refrain from cross-border travel and avoid unnecessary trips outside during the state of emergency [16]. The quasi-emergency measure was the second major restriction after declaring a state of emergency (S1 Fig.). In Japan, a significant reduction in inter-prefectural travel was achieved even without the major restriction by the government between the pandemic periods [17]. We confirmed the appropriateness of the present study period.

Supporting Information (Line 622-628)

S1 Fig. Indices of tertiary industry activity (base year is 2015 and is set at 100) and the monthly number of newly confirmed COVID-19 cases. 

a) Indices of travel agency activity. b) Indices of the passenger transportation industry activity. The declaration of state of emergency was implemented from April 7 to May 25, 2020; from January 8 to March 21, 2021; from April 25 to June 20, 2021; and from July 12 to September 30, 2021. The quasi-emergency measure was implemented from April 5 to September 30, 2021; and from January 9 to March 1, 2022.

References

7. Sharma H, Ilyas A, Chowdhury A, Poddar NK, Chaudhary AA, Shilbayeh SAR, et al. Does COVID-19 lockdowns have impacted on global dengue burden? A special focus to India. BMC Public Health 2022;22: 1402. doi: 10.1186/s12889-022-13720-w

15. Ministry of Economy, Trade and Industry.Monthly reports of Indices of Tertiary Industry Activity. [Cited 2023 Jan 5]. Available from: https://www.meti.go.jp/english/statistics/tyo/sanzi/index.html

16. Cabinet secretariat. COVID-19 Information and Resouces. [Cited 2023 Jan 5]. Available from: https://corona.go.jp/en/emergency/

17. Hara Y, Yamaguchi H. Japanese travel behavior trends and change under COVID-19 state-of-emergency declaration: Nationwide observation by mobile phone location data. Transp Res Interdiscip Perspect. 2021;9: 100288. doi: 10.1016/j.trip.2020.100288.

C3.Covid may have also disrupted surveillance efforts and decreased access to healthcare, leading to a change in reporting rates. This effect could significantly influence your results, especially the static comparisons between pre and post-pandemic disease burden. This should be investigated directly (perhaps by comparing metrics that could serve as proxies for reporting rate, like test positivity and non-Covid medical appointments in the pre and post-pandemic periods) as it could significantly impact the study’s findings.

(Response 3) We appreciate your comment. The decline in access to healthcare was most pronounced in our country in April and May of 2020, when the first state of emergency for COVID-19 pandemic was declared. After the lifting of the emergency declaration, the decrease in access to healthcare was eliminated, but not completely. The reduction rate of access to healthcare varied by department. A surrogate indicator of access to healthcare is the monthly number of people who received health insurance treatment at a primary health care provider's clinic. Each medical facility submits monthly health insurance claims to the Health insurance claims review & Reimbursement Services for calculation of medical costs. Health insurance claims review & Reimbursement Services analyzes the health insurance claims in aggregate. The aggregate results are reported in the monthly statistical report. Therefore, we collected data on the number of health insurance claims from each monthly statistical report. However, we were unable to remove the confounding factor of refraining of access to healthcare from the incidence of each vector-borne diseases. We therefore described it in the materials and methods section and discussed it in the discussion section.

 Material and Methods

 Datasets on access to healthcare (Line 160-164)

We collected data on the number of health insurance claims from monthly statistical reports as surrogate indicator of access to healthcare. The data collected from monthly statistical reports were published by the Health insurance claims review & Reimbursement Services [24]. Those data are presented in the S6 Table.

Results

COVID-19 incidence and access to healthcare (Line 221-228 )

We examined restrictions in access to healthcare during the COVID-19 pandemic (S2 Figure). Figure S2 shows the monthly number of patients who visited the clinic (19 beds or less) as outpatients with health insurance. The value shows the percentage change in the same month in 2019 as 100. The departments concerned with the examination of VBDs were internal medicine, pediatrics, and dermatology. The total number of outpatients was at its lowest from April to May 2020 and in January 2021, when the state of emergency was declared; however, there was a rapid recovery trend the following month. The dynamics differed by department. Pediatrics and internal medicine showed large declines, whereas dermatology showed a smaller decline.

S2. Figure. Number of receipts by department for medical clinics compared with the same month in 2019.

Discussion (Line 404-418)

This study showed the decline in access to healthcare during the COVID-19 pandemic period. Did this not affect the incidence of VBDs in this study? Certainly, internal medicine and pediatrics showed a large refrain rate; however, the reduction in dermatology visits was less in this study. In addition, there may be cases of refusal or postponement of consultations by medical personnels. Therefore, it was difficult to calculate the actual number of cases of refusal to visit a healthcare center and to eliminate the impact of such refusal. However, the major mosquito-borne infectious diseases, dengue fever, Chikungunya fever, and Zika fever if manifested, are characterized by fever and a generalized rash [53]. These infections are difficult to distinguish by clinical symptoms. For tick-borne diseases, scrub typhus (n=4,185), and JSF (n=1,765), fever was observed in 95% and 99% of cases and rash in 86% and 94% of cases [54]. The non-specific fever in the early stages of these diseases may resemble those of COVID-19, making it more difficult for patients to distinguish between them. In addition, as shown in this study, the majority of patients with tick-borne diseases were elderly persons. Therefore, it is likely that the number of visits to the pediatric department is quite low, even during the non-pandemic period. In endemic areas, a high proportion of visits to dermatologists were considered to be made if there was a history of tick bites. Therefore, we　concluded that the effect of avoiding visiting a healthcare center was relatively small.

References

24. Health insurance claims review & Reimbursement Services, Monthly statistics reports. [Cited 2022 Decmber 30]. Available from: https://www.ssk.or.jp/tokeijoho/geppo/index.html

53. Huntington MK, Allison J, Nair D. Emerging Vector-Borne Diseases. Am Fam Physician. 2017 Jun 15;95: 758. PMID: 27929218.

54 Kinoshita H, Arima Y, Shigematsu M, Sunagawa T, Saijo M, Oishi K, et al. Descriptive epidemiology of rickettsial infections in Japan: Scrub typhus and Japanese spotted fever, 2007-2016. Int J Infect Dis. 202;105:560-566. doi: 10.1016/j.ijid.2021.02.069. 

Supplemental Information (Line 629-633)

S2 Figure. Number of receipts by department for medical clinics compared to the same month in 2019. Figures indicate the number of patients visited outpatient of the clinic by month. The declaration of state of emergencywas in effect from April 7 to May 25, 2020; from January 8 to March 21, 2021; from April 25 to June 20, 2021; and from July 12 to September 30, 2021. In addition, quasi-emergency measures were in effect from April 5 to September 30, 2021.

C4.The Olympic Games are mentioned several times, but it’s unclear what the hypothesized effect is and what VBD and Covid precautions (if any) were taken while the games happened. It may even be worth considering whether the Olympic Games themselves constitute another event that may have separately altered VBD transmission. Line 356 states that there was no “no significant increase” during the Olympic Games, but doesn’t include an analysis to support this claim.

(Response 4) Thank you for your comment. The impact of the Tokyo Olympics is not clear from this study. The only thing that was clear is that the number of people coming to Japan from Europe increased during the Olympic Games. There was no evidence that this was related to the Olympics. Moreover,, at the Tokyo Olympics, transportation from the athletes' village to the venue was by bus, and outings and personal activities were restricted. Furthermore, because the games were held without spectators, there was no general visitor immigration or assembly from within the country. Therefore, we believed that the impact on the occurrence of VBDs in the country was minimal. Therefore, in the Results section, we only presented the duration of the Olympic Games, and in the Discussion section, we only discussed the fact that the number of visitors from Europe increased during the period of the Olympics rather than the Olympics, and that this poses little risk of VBDs being imported into Japan from the present results. This was reflected in the discussion below, and redundant language regarding the Olympics has been removed as follow:

.Introduction

Old line 65-67: More than two years have passed since the COVID-19 pandemic. During that time, Japan has hosted international events such as the Olympic Games, and the amount of human mobility has changed significantly.

Old line 194-197: We note that the Tokyo 2020 Olympic and Paralympic Games (Olympics Games Tokyo 2020) were held in Japan in the summer season of 2021. The Olympic Games were held during the 'fifth wave' of COVID-19 (weeks 30–32 in 2021) and the Paralympic Games were hosted from weeks 35–36 in 2021, with no spectators other than the athletes themselves and any necessary staff (Fig. 2).

Old line 356-359: There was no significant increase in the number of new imported cases during the 2020 Olympic Games in Tokyo (held in 2021). We attribute this to restrictions on congestion and mobility during the Olympic Games. Therefore, the impact of human mobility during the 2020 Olympic Games on imported vector-borne diseases was probably small. 

Old line 361-362: Since this occurrence was only noted during the 2020 Olympics Games, we assume that this fluctuation was associated with hosting the games. 

C5. In general, the manuscript considers several diseases and covariates, which can make the paper difficult to follow. Better organization throughout and a schematic figure that illustrates the key questions/proposed mechanisms could help to guide the reader.

(Response 5) Thank you for your valuable comments and suggestions. First, the figures and tables were organized. Those of high importance were made into figures and tables, whereas those of low importance were made into supporting figures. For tables, some tables were merged, although a new table for statistical analysis was added. Specifically, the regional incidence of vector-borne diseases is summarized in one table. The sex and age-specific proportions of domestic infections are summarized in one table. We also organized subtitles of the section of results. Several representative covariates affecting the incidence of diseases were organized and illustrated in a figure. 

Discussion (Line 419-437)

As we have discussed, there were a variety of correlated factors for the incidence of VBDs [Fig. 4]. Human mobility appeared to have more impact on mosquito-borne infectious diseases but little impact on tick-borne diseases. Instead, geographical or seasonal factors had a greater impact for tick-vector diseases. Commodity distribution was considered to have a small impact on both diseases. Age and sex were influenced by other confounding factors, which may secondarily affect infection rates. For example, as shown in this study, differences in the proportion of agriculture and forestry workers by age group may affect the incidence of tick-borne diseases. Sex differences were not shown in this study, either before or after the epidemic; however, if there were differences in certain behaviors between men and women, it may have an impact. Other correlating factors were not considered in this study, although, it should be noted that the influence of each correlated factor on the incidence of mosquito- and tick-borne diseases varies between countries and regions. For example in countries with distinct wet and dry seasons, climatic factors may have a strong influence on mosquito-borne disease outbreaks beyond seasonal effects [56]. Moreover, animals with ticks move into residential areas, expanding the distribution area of ticks, especially raccoon migrations, which can affect the distribution of various ticks [57].

Fig. 4. Various correlating factors influencing the incidence of vector-borne diseases. 

a) Imported/mosquito-vector borne infectious diseases. b) Endemic/tick-borne diseases. Each ring represents a correlating factor, each of which relates to each other and influences the incidence of vector-borne diseases.

References

56. Zhou G, Minakawa N, Githeko AK, Yan G. Association between climate variability and malaria epidemics in the East African highlands Proc Natl Acad Sci U S A. 2004;101: 2375-2380. 

57. Doi K, Kato T, Tabata I, Hayama S-I. Mapping the Potential Distribution of Ticks in the Western Kanto region, Japan: Predictions Based on Land-Use, Climate, and Wildlife. Insects. 2021;12: 1095. 

Minor comment:

Below, I mentioned a few cases where analyses and their corresponding hypotheses are not in the introduction, as well as strong causal claims that are made without sufficient evidence. Such claims should be supported with either additional analyses or references to other studies.

We have responded to each of your comments, taking note of the points you raised, and have provided references where necessary.

C1. Abstract

a. line 19: “common” should be “domestic” or “endemic” (and be consistent throughout)

(Response1a) We have revised it as follows:

Line 20-21: …..and resulted in the incidence of both endemic and imported infectious diseases.

b. line 22: clarify that the analysis was conducted using data for Japan at the regional level

(Response 1b) The following sentence was added to the text as follow: 

Line 23-24: …The analysis was conducted using data through the National Epidemiological Surveillance of Infectious Diseases system in Japan.

c. line 33: change semicolon to period

(Response 1c) Finally, the following revisions were made

Line 33-34: The incidence of severe fever with thrombocytopenia syndrome, scrub typhus, and Japanese spotted fever did not show changes between the two periods and either no association with human mobility. 

C2. Introduction

a. line 44: were there restrictions on domestic travel? Please also include discussion of when restrictions were implemented and how long they were in place

(Response 2a) In Japan, relatively moderate nationwide restrictions on behavior took place under a state of emergency declaration. This state of emergency was declared four times from 2020 to 2021. The details of the periods are shown in S1/S2 Figure. We mentioned this in the introduction section.

Line 45-47: ….In Japan, restrictions were placed on entry from endemic countries, and during the nationwide declaration of a state of emergency, restrictions were placed on movement out of prefectures. 

　　　　 

S1 Fig. Indices of tertiary industry activity (base year is 2015 and is set at 100) and the monthly number of newly confirmed COVID-19 cases. (Line 622-628)

a) Indices of travel agency activity. b) Indices of the passenger transportation industry activity. The declaration of state of emergency was declared from April 7 to May 25, 2020; from January 8 to March 21, 2021; from April 25 to June 20, 2021; and from July 12 to September 30, 2021. In addition, quasi-emergency measures were declared from April 5 to September 30, 2021; and from January 9 to March 1, 2022. 

b. line 45: remove “unnecessary”

(Response 2b) We revised it as follow:

Line 43-44: ….gatherings, and the implementation of unnecessary curfews [4]

c. line ~ 49: You should also mention that Covid restrictions could lead to increases in vector-borne diseases by disrupting medical services and vector control campaigns.

(Response2c) Thank you for your suggestion. We have mentioned the points you raised in the section of discussion as one possible explanation for the increase in tick-borne diseases.

Line 401-403: Moreover, we cannot deny that mobility restrictions for the COVID-19 pandemic could lead to increases in VBDs by disrupting medical services and vector control campaigns [52].

References

52. Reegan AD, Gandhi MR, Asharaja AC, Devi C, Shanthakumar SP. COVID-19 lockdown: impact assessment on Aedes larval indices, breeding habitats, effects on vector control programme and prevention of dengue outbreaks. Heliyon. 2020;6: e05181. doi: 10.1016/j.heliyon.2020.e05181.

d. Line 59: mention Covid impacts on trade here

(Response 2d) Thank you for your suggestion. I have made the following additions:

Line 60-61: Nevertheless, it is unclear whether the increase or decrease in foreign and domestic trade has an effect on the incidence of domestic VBDs.

e. Line 63: explain why you would expect a difference between short and long-term effects

(Response 2e) During the 3 years of life in the corona fossa, strictly speaking, each region had different periods of behavioral inhibition. Therefore, the most significant results could have been obtained if the study period had been set in a short period of time when behavioral inhibition was strongest. However, the countries where non-coercive behavioral inhibition took place showed small undulating mobility changes over the 3-year period. Therefore, we considered it necessary to look at the long-term effects to see the impact of this policy on the epidemic of VBDs. For this reason, we used the term long-term effects; however, we have deleted the sentence to simplify the text.

f. Line 65: “since the COVID-19 pandemic” – arguably, the pandemic is ongoing, so it may make more sense to say “since the COVID-19 pandemic began”

(Response2f) Thank you for your important suggestion. However, we have deleted this sentence when we deleted the descriptions about Olympic games. However, we will use that expression in the following paper.

g. In the introduction, you should also discuss domestic mobility and where humans tend to come into contact with the different types of vectors to develop hypotheses for how diseases may have changed during the pandemic based on mobility changes.

(Response2g) We described it as follows: 

Line 62-67: Tick-borne pathogens are maintained in enzootic cycles involving ticks and wild animal hosts, with epizootic spread to other mammals, including livestock and humans [10]. Therefore, natural habitats of animals, especially wild beasts and birds, may harbor many ticks. On the other hand, malaria and dengue fever are diseases transmitted through mosquitoes to humans/animals. Pathogen-bearing mosquitoes can be found around humans dwelling in urban areas [11]. Therefore, the incidence of VBDs may be affected by the movement of people in urban as well as rural areas.

References

10. Yamaji K, Aonuma H, Kanuka H. Distribution of tick-borne diseases in Japan: Past patterns and implications for the future. J Infect Chemother. 2018;24:499-504. doi: 10.1016/j.jiac.2018.03.012.

11. Higa Y. Dengue Vectors and their Spatial Distribution.Trop Med Health. 2011 Dec;39(4 Suppl):17-27. doi: 10.2149/tmh.2011-S04.

h. In the introduction, you should introduce the ideas behind the estimated infected areas analysis, gender, and age-specific incidence analysis, why you think the distribution may have changed under Covid, and the hypothesized direction of that change.

(Response2h) We have discussed this in the Introduction section as follow:

Introduction

Line 68-73: It is unclear whether the sex- and age-specific incidence rates of VBDs changed during the COVID-19 pandemic. A study conducted during the pandemic noted a strong association between sex and COVID-19-related fear and anxiety [12]. Fear and anxiety may discourage outgoing behavior [13]. Moreover, avoidance of social and mass gatherings was shown to change with age [14]. Hence, there may also be sex- and age-related variation in the incidence of VBDs between the pre-pandemic period and during the COVID-19 pandemic.

References

12. Metin A, Erbiçer ES, Şen S, Çetinkaya A. Gender and COVID-19 related fear and anxiety: A meta-analysis. J Affect Disord. 2022;310: 384-395. doi: 10.1016/j.jad.2022.05.036. 

14. Korn, L., Siegers, R., Eitze, S., Sprengholz, P., Taubert, F., Böhm, R. et al. Age differences in COVID-19 preventive behavior: A psychological perspective. European Psychologist. 2021;26: 359–372. doi: 10.1027/1016-9040/a000462.

i. Throughout the manuscript, you use the term “mobility restriction” generally, but it would be useful to clarify when you’re discussing bans on international travel vs reduction in local travel, as the expected outcomes differ.

(Response2i) Where the description includes both meanings, it has been left as is, but where it can be clearly stated, the respective sections have been revised as follows.

Line 22-23: Hence, we aimed to investigate the impact of transborder and local mobility restriction on vector-borne diseases

Line 51-52: Non-mandatory local mobility restrictions implemented in Taiwan and Japan may exhibit different effects with regard to domestic VBDs epidemics.

Line 364-365: This would suggest that tick-borne diseases were not affected by changes in local mobility. 

Line 401-403: Moreover, we cannot deny that local mobility restrictions for the COVID-19 pandemic could lead to increases in VBDs by disrupting medical services and vector control campaigns [52].

Line 461: ...however, transborder mobility restrictions may influence the incidence,

C3. Material and Methods

a. Line 122 – what percent of people riding bullet trains are commuter (are you excluding a significant proportion of train riders from your analysis?)

(Response 3a) The percentage of commuter pass users using bullet train was 13.8% (51 million/370 million) in 2019, 26.9% (42 million/156 million) in 2020, and 21.5% (42 million/195 million) in 2021. In other words, about 80% of the passengers were non-scheduled users during normal period. In this analysis, we excluded those who used the bullet train with a commuter pass. The reason was that many of these individuals are commuters, who are unlikely to engage in behaviors that would put them at high risk for transmission of vector-borne diseases. Additional explanations are provided in the Material and Methods section as follows:

Line 129-131: The number of railway passengers accounted only for people who boarded the bullet train, excluding commuter pass users. This is because most commuter pass users were unlikely to visit vector habitats.

b. Line 131-136 – is there precedent for using passenger-kilometer and ton-kilometer in this manner as metrics for passenger/cargo transport? Please cite other studies that use them.

(Response 3b) No appropriate literature could be found. We intended this study to focus on long-distance travel for sightseeing and outdoor activities rather than travel within the daily living area. Therefore, we employed the number of passenger kilometers rather than the number of passengers. We have added additional explanations to the section of Materials and Methodas follows:

Line 145-148: …We intended this study to focus on long-distance travel for sightseeing and outdoor activities, rather than travel within the daily living area. Therefore, we employed the amount of passenger-kilometers rather than passenger number. The ton-kilometer was chosen to match passenger kilometers.

c. Line 150 – “estimated regions of infection” please explain in greater detail what this means and how it was determined. Is it based on individual follow-up for each case?

Line 159 – you should also mention in this paragraph that you have data on whether particular cases of each disease were transmitted locally or imported and explain how that information was initially assessed.

(Response 3c) Reply to reviewer: The estimated region of the infection was determined by physician interviews for individual patients. This was added to Materials and Methods as follows:

 Materials and Methods

Line 172-175: Data on incidence categorized by suspected region of infection, age group, and sex were also obtained [27]. These information obtained were queried in standard physician interviews. As for the estimated region of infection, physicians extrapolated these details from the patient's travel history, location of the insect bite, and other details.

d.Line 174 – clarify that you are comparing annual burden in either period (otherwise it’s confusing to compare a 4 vs 2 year period)

(Response 3d) Reply to reviewer: We have deleted the description for t-test. This was due to the mixed results of the ITSA and the t-test in the text.

e. Line 177-179 – is this part of the analysis described in the previous paragraph? If so, make that clear

(Response 3e) We have deleted this description.

C4. Results

a. Line 205 – clarify it was “on the rise” prior to the pandemic’s beginning

(Response 4a) Reply to reviewer: We revised it as follow:

Line 230-231: The number of foreigners entering Japan had been on the rise prior to when the COVID-19 pandemic began (S3a Fig).

b. Line 205-235 – a lot of the information given here feels extraneous, which makes it difficult to keep track of the general trends that you’re describing in this section

(Response 4b) We revised these paragraphs as follow: 

 International and domestic movements (Line 230-244):

The number of foreigners entering Japan had been on the rise prior to when the COVID-19 pandemic began (S3a Fig). The number of Asian foreigners increased yearly. In December 2019, the proportion of Asian foreign nationals entering the country was 82.8%, whereas that of non-Asians was 17.2%. In February 2020 (at the start of the global COVID-19 pandemic), the number of immigrants began to decline. Although the ratio of visitors from Asia remained high despite this decline, the ratio of visitors from Asia declined and the ratio of visitors from Europe increased only in July and August of 2021(S3a Fig. inset). Since 2016, the number of Japanese returnees had remained above 1 million each month, although there had been monthly fluctuations (S3b Fig). However, this number has dropped to 10,000 by April, 2020 and was less than 5% of the total returnees in 2019. The number of containerized cargo handled in external trade during the pandemic declined at a rate of only -2.3% during the pandemic compared to the pre-pandemic periods (S3c Fig). The mobility of people within the country during the pandemic period showed a decrease within each form of transportation service. Person-kilometers decreased 58.3% by automobile, 63.5% by railway, and 55.1% by air transportation compared to the pre-pandemic period (S4 Table). Trends in domestic cargo during the pandemic period showed a small decrease within each of the transportation services, except aircraft (S4 Table).

c.Fig 3 – add titles and y-axis labels to each of the plots to make it easier to understand what each panel is plotting

All tables – note any significant differences in bold

(Response 4c) We have added titles to each graph. The position of the label on the y-axis has been changed. Note that Fig. 3 has been moved to S3 Fig.

d. Line 250 – “an overall decline” -> “fewer cases”

(Response 4d) Thank you for your comment. We have deleted this sentence as follows:

Old line 249-250: For dengue fever, seasonality was obscured and an overall decline was observed in the pandemic period.

 ➡

Line 253-254: For both diseases, apparent seasonality was obscured during the pandemic period. 

e. Line 256 – “statistically” -> “statistical”

(Response 4e) We have revised it as follow:

 Line 269: We conducted ITSA as statistical analysis…

f. Line 256-258 – “malaria showed significantly…Japanese spotted fever” need more detail here. What were the estimated slopes and intercepts? It also seems like you’re missing discussion of dengue and Japanese spotted fever.

(Response 4f) We have revised this sentence as follow:

Results (Line 269-281)➡ For more information, see “Response 1”.

g. Lines 282 and throughout – try to avoid using “significant” if you’re not referring to the result of a statistical analysis

(Response 4g) We have revised it as follows:

Line 299-300: Relapsing fever was confined to Hokkaido during the pandemic, although endemics were seen in areas outside of Hokkaido before the pandemic (Table 3).

h. Fig 4 – It’s inconsistent to have an average for 2016-2019 and then the yearly values for 2020 and 2021. Can you show just individual years or just averages for all of them (although that may get a bit crowded)? I also think you can move some of the panels to the supplement and just focus on a few of the disease that showed the most interesting/significant trends with this figure.

(Response 4h) We have shown epidemiological curves for individual years as a trial; however, some diseases are very complicated. Therefore, we showed the epidemic curve of individual years in S4 fig. Moreover, we focused on a few of the disease that showed the most interesting/significant trends with the figure.

S4 Fig. Current epidemic curves for vector-borne diseases since 2016 in Japan. SFTS: severe fever with thrombocytopenia syndrome.

i. Table 3 - for pre-pandemic SFTS in West Japan, it looks like the percentage is wrong (should be 99.7)

(Response 4i) My apologies, you are correct. We have corrected it to 99.7%. See line 298.

C5. Discussion

a. Line 320 – “was suppressed” I don’t think you actually showed this statistically. If so, please reference the relevant analysis. I would also change the order of this to lead with your most important findings.

(Response 5a) We have revised this paragraph as follows:

Line 327-333: This study presented data on the incidence of VBDs under conditions of restriction of human mobility during the COVID-19 pandemic. We found that the onset of malaria and dengue fever showed significant decrease in the first year of the pandemic, and those may be positively correlated with changes in the number of arrivals to the country. On the other hand, there was no significant change in the incidence of JSF, SFTS and scrub typhus during the pre-pandemic and pandemic periods. Their epidemiological curves showed more seasonal pattern rather than correlating with behavioral changes in the domestic population.

b. Line 322 – “Maralia” -> “malaria”

(Response 5b) Thank you for your detailed review. We have revised “Maralia” to “malaria” throughout the text.

 .

c. Line 339 – “was only 0.87% of that in 2019” this is quite low, so make sure the number is correct and also cite your source/the table you’re referencing for this.

(Response 5c) Reply to reviewer: We have revised this sentence as follows:

Line 348-350: However, the number of immigrants from Asia to Japan during pandemic periods declined by -92% compared to the pre-pandemic period.

d. Line 347-355 This paragraph is confusing. What do you mean by saying trade is “currently saturated and unlikely to increase/decrease?” How does it compare to the pre-pandemic baseline or the time period observed in the study? “A rapid epidemic would have occurred…” is an extremely strong claim, and there are no analyses or sources here to support it.

(Response 5d) Thank you for your observation. We have revised this paragraph as follow:

Line: 421-422: ….Commodity distribution was considered to have a small impact on both diseases. However, mosquitoes can be spread around other areas of the world by ship transport [55]. 

e. Line 357 – “We attribute this” you don’t have sufficient evidence to make this causal claim

(Response 5e) Reply to reviewer: Thank you for this insight. We have deleted the sentence “We attribute this”.

f. Line 363 – “However, the number of patients…” -> “However, a change in the number of patients”

(Response 5f) This sentence has been revised as follow by correcting the entire paragraph:

Line 355-356: Additionally, no VBDs cases were brought in from North America or Europe throughout the study period, although the percentage of visitors from Europe increased, surpassing visitors from Asian countries in July 2021.

g.Line 367 – “travel for tourism purposes” explain more why you think this would have changed the distribution of travel and provide sources.

(Response 5g) The percentage of Japanese travelling abroad by purpose was published annually; however, during the pandemic period, the number of data was small and not published. For this reason, the sentence has been deleted for lack of supporting data.

Old line 366-368: This is thought to have occurred because travel for tourism purposes was restricted while travel for business purposes was allowed; however, no vector-borne…..

h. Line 378-383 – too much detail provided here about someone else’s study

(Response 5h) We have revised it as follows.

Line 368-371: A study of SFTS cases in Japan reported that the most frequent activity at the time of infection was farming (65%) [41]. A national study of scrub typhus in Japan showed that the activity engaged in at the time of infection was acquired included farming (31.6%) and forestry work (14.2%) [42]. Another study of JSF reported that farming (35.7%) was the most frequent activity [43]. 

i.Line 398 – “ these may be plausible” please try to find additional sources to support this hypothesis. Have other people found similar trends?

(Response 5i) Thank you for your suggestion. Similar considerations are likely to be possible in South Korea. The median age of patients with SFTS in South Korea is also 69 years, many of whom are infected during agricultural work [44]. And 68.3% of agricultural workers were over 60 years of age [45]. This suggested that the ageing of the farming population may be linked to the high incidence of SFTS among elderly people. But we have discussed the following, citing Korean studies: 

Discussion

Line 375-379: In the East Asia, the elderly group aged 60 years or older was the most common age group in patients with SFTS, scrub typhus, and JSF [41-44]. Similarly, in our study, most patients with these diseases were in the elderly group throughout the two study periods. The average age of Japanese farmers was 67.8 years in 2020 and 69.6% of farmers were aged ≥65 years in 2020, making this age group the core of independent farm management in Japan, which was similar to situation in Korea [45],….

References

44. Shin J, Kwon D, Youn SK, Park JH. Characteristics and Factors Associated with Death among Patients Hospitalized for Severe Fever with Thrombocytopenia Syndrome, South Korea, 2013. Emerg Infect Dis. 2015;21: 1704-10. doi: 10.3201/eid2110.141928.

45. Statics Korea. Census of Agriculture, Forestry and Fisheries.[Cited 2022 December 31]. Available from: https://affcensus.go.kr/eng/mainView.do

j. Line 404 – “ humans are not an infectious source” – what does this mean? Humans can’t transmit back to the vector?

(Response 5j) I apologize for my lack of explanation. We have revised it as follows:

Line 385-387: For scrub typhus and JSF, the pathogens were transmitted between vectors via gametes rather than through ticks sucking blood from an infected person or animal [47].

k. Line 406 – “suspected region of infection” relate this back to a specific table and cite it

(Response 5k) Before line 406, the suspected region of infection was described in association with Table 3 as follows.

Line 384-385: This study showed no significant changes in the suspected region of infection: SFTS and JSF were dominant in Western Japan, Scrub typhus was dominant in Eastern and Western Japan.

l. Line 423 – “various studies” -> cite the studies you’re referencing

(Response 5l) Thank you for pointing this out. However, to make the discussion shorter, this section has been deleted.

Old line 422-424: The high prevalence in young people and in males seen in various studies may reflect the characteristics of Borrelia spp., a pathogen of both diseases.

m. Line 443 – “new residents in rural areas” - do regional patterns bear this out? What other data and tables from the study can you cite to support this?

(Response 5m): We apologize for the lack of explanation in the cited reference (37); Fielding et al. reported an increase in migration to rural areas away from metropolitan areas in Japan, using inter-prefectural population movement data. We have added this to the text. Their data source is the Report on Internal Migration in Japan published by the Statistics Bureau of the Ministry of Internal Affairs and Communications. 

Discussion: line 397-398: Fielding et al. reported an increase in migration to rural areas away from metropolitan areas in Japan under the pandemic, using inter-prefectural population movement data [57]. This was thought to reflect the growing interest in moving to and working in rural areas to avoid living in densely populated regions during the coronavirus pandemic [51]. 

n.Line 461 – “new infectious diseases such as COVID-19” this sentence is confusing because Covid isn't a vector-borne disease

(Response 5n) Thank you for your comment. However, we have deleted the section to shorten the text.

Old line 460-461: Therefore, in the post-COVID-19 era, Japan will continue to import vector-borne diseases, thus posing a risk of bringing in new infectious diseases such as COVID-19.

 

Reviewer #2:

In this study, the authors examined the impact of mobility restriction on vector-borne diseases during the COVID-19 pandemic in Japan through a descriptive epidemiological study. They collected both mobility data and several typical vector-borne diseases data. They performed a simple interrupted time series analysis. I have a few comments:

We would like to thank you for your suggestion and insightful comments. We have replied to each of your comments as follows:

C1.In line 41, “Wuhan (Zhejiang Province, People's Republic of China)”. This might be corrected as “Wuhan (Hubei Province, People's Republic of China)”.

(Response 1) We have revised “Zhejiang Province” to “Hubei province” as follows:

Line 39-40: In December 2019, the coronavirus disease 2019 (COVID-19) pandemic began with an outbreak in Wuhan (Hubei Province, People's Republic of China).

C2. Vector-borne diseases have a strong seasonality and subject to the level of population immunity gained through previous infections [ref 24]. The authors may wish to account for these confounding factors. Otherwise, it is difficult to understand if the observed pattern in incidence is due to restrictions or other factors.

(Response 2) Thank you for your important comment. In the preliminary study, more advanced statistical methods such as autoregressive integrated moving average (ARIMA) and seasonal-ARIMA (SARIMA) could be considered; we had tried ARIMA, but the predictive epidemic curve changed depending on whether the period before the pandemic was set in 2019 or from 2016 to 2019, which makes it arbitrary. SARIMA, which eliminates seasonality, could not be performed this time because the statistical analysis software could not accommodate it. We used interrupted time-series analysis (ITSA) for this statistical analysis because we considered that the main purpose of this analysis was to show "changes" or "differences" that deviated from the trend. Another reason for not eliminating seasonality is that this study was based on national data. Japan is made up of islands of various sizes, and its length, from north to south, is over 2,000 km. The country's four distinct seasons feature three periods of heavy precipitation. These three wet periods shove the nation's average annual precipitation which is almost double of the world average. Due to the large differences in topography, climatic variations, and climate in the region, we considered that it would be impossible to analyze the data using national averages. These have a significant impact on the habitat range of vector species. For example, in scrub typhus, the epidemics occurred throughout the country, but the peak of the epidemic in eastern Japan was in the summer, whereas the peak in western Japan was in winter [IASR]. Such seasonality also differs from region to region and, therefore, must be analyzed by region, but this could not be done in the present study due to limitations in data collection. This was briefly described in the discussion as follows:

Discussion

Study limitation (Line 439-456)

VBDs have a distinct seasonality, as shown in this study. It was, therefore, important to eliminate the influence of seasonality. Japan is made up of islands of various sizes, and its length, from north to south, is over 2,000 km. Japan is predominantly mountainous [58]. The Japan Alps, studded with 3,000-meter peaks, bisect the central portion of the main island [58]. The climate is not uniform, ranging from subtropical in the south to subarctic in the north. The presence of the Blakiston’s and Itoigawa-Shizuoka Tectonic Lines, which divide the insects-living areas, is also evidence of the diversity of climate. For example, in scrub typhus, the epidemics occur throughout the country, but the peak of the epidemic varied by region because of regional differences in its vector species [54]. Therefore, it was difficult to eliminate seasonality or climatic correlating factors in the present study using national data. In the future, we intend to examine the influence of climate in a regionally restricted study.

The onset of VBDs was subject to the level of population immunity gained through previous infections. However, this study was a national survey using a national data base and antibody levels in the population were not available; therefore, the impact of this could not be excluded. However, there was a study that examined antibody levels in local populations in an endemic area for SFTS [59]. In this study, one out of 694 individuals tested positive for antibody titers. This meant that even in endemic areas, the local population was unlikely to be immune to SFTS. The rate of rickettsia-bearing toxic ticks in scrub typhus (Leptotrombidium sp.) was estimated to be 0.1–3% [53]. The percentage of antibodies to scrub typhus was, therefore, considered to be quite low. However, this is an issue to be considered in the future.

References

54. Kinoshita H, Arima Y, Shigematsu M, Sunagawa T, Saijo M, Oishi K, et al. Descriptive epidemiology of rickettsial infections in Japan: Scrub typhus and Japanese spotted fever, 2007-2016. Int J Infect Dis. 2021;105: 560-566. doi: 10.1016/j.ijid.2021.02.069. 

58. Ministry of Land, Infrastructure, Transport and Tourism, Land and Climate of Japan. [Cited 2022 December 30]. Availablefrom:https://www.mlit.go.jp/river/basic_info/english/land.html

59. Kimura T, Fukuma A, Shimojima M, Yamashita Y, Mizota F, Yamashita M, et al. Seroprevalence of severe fever with thrombocytopenia syndrome (SFTS) virus antibodies in humans and animals in Ehime prefecture, Japan, an endemic region of SFTS. J Infect Chemother. 2018;24: 802-806. doi: 10.1016/j.jiac.2018.06.007. 

C3. In line 174, “estimated number of infections”. Could you explain how the number of true infections was estimated in this study?

(Response 3) The confirmation of infection is made by a doctor based on diagnostic criteria such as PCR and antibody tests, together with symptoms and blood tests. The diagnosing doctor reports all cases to the local health center based on the infectious disease surveillance system, and this information is then collated by the National Institute of Infectious Diseases. The text does not mention the details of the process but we additionally described that the incidences are determined on the basis of the infectious disease surveillance system and that information such as the suspected infection area is obtained from the doctor's interview.

Old line 174-176: We used Student’s t-tests to compare differences in the estimated number of infections by geographic region, age, and sex between the pre-pandemic period (2016–2019) and the pandemic period (2020–2021). All p-values were two-sided, and p<0.05 was considered statistically significant.

Line 23-25: The analysis was conducted using data through the National Epidemiological Surveillance of Infectious Diseases system in Japan.

Lin 166:170: In this study, VBDs included all vector-borne diseases among the nationally notifiable diseases within the National Epidemiological Surveillance of Infectious Diseases (NESID) system operated by the National Institute of Infectious Diseases and the MLHW. The reported number of infectious disease cases was collected from the Infectious Diseases Weekly Reports published by the National Institute of Infectious Diseases [25].

Line 172-175: Data on incidence categorized by suspected region of infection, age group, and sex were also obtained [27].These information obtained was queried in standard physician interviews. As for the estimated region of infection, physicians extrapolated these details from the patient's travel history, location of the insect bite, and other details.

Reviewer #3: 

This manuscript, “Effect of human mobility restriction on vector-borne diseases during the COVID-19 pandemic in Japan”, provides a nice summary of how the incidence of imported and domestic vector-borne disease cases have shifted during the COVID-19 pandemic period in pandemic compared to the pre-pandemic period. The analysis includes all vector-borne diseases that are nationally notifiable, which allows the authors to look at broader trends that might be missed in an analysis of a single pathogen. The manuscript is labeled as a descriptive epidemiological study, however,I believe that the statistical analyses that are conducted need improvement. In addition, some of the results need to be more clearly presented as in some sections of the Results and Discussion it is stated that several of the domestic diseases increased during the pandemic, but then in another section it will say that the same diseases did not experience a statistically significant change in incidence. If this is due to differences in the results of the different statistical tests (t-test vs. ITSA) then which test is being referred to needs to be clearly stated.

We appreciate your valuable comments. Regarding ITSA, we have re-analyzed the data using monthly data. As the t-test and ITSA were mixed in the text, we have removed the t-test with respect to statistical analysis for the incidence of vector-borne diseases. Details were provided in the replies to individual comments.

Major comments:

C1. No values are reported for any of the statistical analyses (e.g., the t-score and p-values for the t-tests, or p-value and coefficient estimates for ITSA). It is also not clear which analysis (the t-test or ITSA) is being referenced in some of the Results (e.g., on lines 241-244), including the appropriate statistical values will make it clearer which test is being mentioned.

(Response 1) The results of the ITSA are summarized in Table 2. The p-values were also shown for the descriptions in the results.

Results (Line 269–281)

We conducted ITSA to examine significant differences between the pre-pandemic and pandemic periods (Table 2, Fig 3). Before the beginning of the COVID-19 pandemic, dengue fever/malaria onset per month did not significantly increase or decrease every month. In the first year of the pandemic (2020), there appeared to be a significant decrease in dengue fever (p=0.003) and malaria (p=0.002) onset per month, followed by a non-significant increase in the annual trend of dengue fever and malaria onset per month. No significant differences were observed for SFTS (p=0.265), scrub typhus (p=0.207), and JSF (p=0.514) during the pre-pandemic and pandemic periods. However, there was a change in slope from 0.021 to 0.016, for SFTS; from 0.115 to 0.511, for scrub typhus; and from 0.062 to 0.268, for JSF.

Table 2.Interrupted time-series analysis of the incidences of vector-borne diseases for pre- and post-COVID-19 pandemic

Diseases Coefficient p-value 95% CI

Dengue fever Slope prior to intervention 0.060 0.322 -0.060 to 0.180

 Immediate change post-intervention -6.225 0.003 -10.310 to -2.140

 Change in slope post-intervention -0.128 0.076 -0.270 to 0.014

Malaria Slope prior to intervention 0.002 0.724 -0.011 to 0.015

 Immediate change post-intervention -0.812 0.002 -1.322 to -0.302

 Change in slope post-intervention 0.011 0.452 -0.017 to 0.039

SFTS Slope prior to intervention 0.021 0.171 -0.009 to 0.052

 Immediate change post-intervention -0.712 0.265 -1.978 to 0.553

 Change in slope post-intervention 0.016 0.745 -0.084 to 0.116

Scrub typhus Slope prior to intervention 0.115 0.539 -0.256 to 0.487

 Immediate change post-intervention -9.177 0.207 -23.541 to 5.187

 Change in slope post-intervention 0.511 0.327 -0.522 to 1.545

JSF

Slope prior to intervention 0.062 0.314 -0.060 to 0.185

 Immediate change post-intervention -2.373 0.514 -9.594 to 4.849

 Change in slope post-intervention 0.268 0.289 -0.232 to 0.768

CI: confidential interval, JSF: Japanese spotted fever, SFTS: severe fever with thrombocytopenia syndrome

(Response 1) Also, the mention of the t-test has been removed from the text. The part of the t-test that was performed and expressed as significant has been revised as follows.

 Material and Methods

 Statistical analyses

Old line 174-176: We used Student’s t-tests to compare differences in the estimated number of infections by geographic region, age, and sex between the pre-pandemic period (2016–2019) and the pandemic period (2020–2021). All p-values were two-sided, and p<0.05 was considered statistically significant.

➡

Line 196–198: The Pearson correlation was used to examine the relationship between entrants to the country with monthly incidences of VBDs or between domestic human mobility obtained from Google mobility data sets with monthly incidence of VBDs from January 2016 to October 2022.

Line 238–240: The number of containerized cargo handled in external trade during the pandemic declined at a rate of only -2.3% compared to the pre-epidemic period (S3c Fig).

Line 328–333: …We found that the onsets of malaria and dengue fever showed significant decrease in the first year of the pandemic. The two incidences may be positively correlated with changes in the number of arrivals to the country. On the other hand, there was no significant change in the incidence of JSF, SFTS and scrub typhus during the pre-pandemic and pandemic periods. Their epidemiological curve showed more seasonal pattern rather than correlating with behavioral changes in the domestic population.

Line 361–362: SFTS, scrub typhus, and JSF showed no significant changes before and after the pandemic following ITSA.

C2. For the Interrupted Time Series Analyses (ITSA), it is not clear whether the results presented are for significant changes in the level or trend pre- and post-pandemic. Both the Methods and Results sections need to specify which effect is being reported. 

(Response 2) ITSA was presented for significant changes in the level of dengue fever and malaria during the pre- and post-pandemic. There were no significant changes for the each trends for any VBDs in both the pre-pandemic and post-pandemic periods. We mentioned it in the section of the Material and Methods and the Results.

Material and Methods

Statistical analysis

Line 199–210: Interrupted time series analysis (ITSA) was conducted using monthly data to clear whether significant changes for the incidences of individual diseases during pre- and post-COVID-19 pandemic following the method described by Linden [30]. ITSA is a method of evaluating the impact of an intervention on an outcome (the causal effect) in terms of the level of change (the change in intercept) or the trend in change (the change in slope) within a time series [30]. The analysis was focused on diseases whose monthly incidence could be statistically analyzed (an average monthly incidence of >1.5 for domestic endemic diseases) or whose international importance was high such as dengue fever, malaria, SFTS, scrub typhus, and Japanese spotted fever (JSF). First, we examined whether there was a linear underlying time trend of the incidences of VBDs during the pre-pandemic and pandemic periods using the Cochran–Armitage test. Ordinary least-squares regression models designed to adjust for autocorrelation were employed for the single-group ITSA, which included data on the pre-pandemic and pandemic periods

Result

Line 269-281: For more information, see “response 1” for Comment 1.

C3. In addition, because there are only four pre-pandemic data points, and only 2 pandemic data points, testing for changes in trends would be statistically inappropriate. There are also doesn’t appear to be a clear scientific reason why you would want to test for a change in the trend between the two time periods. The Methods section should also mention that the ITSA was conducted using annual incidence (as opposed to the t-test which mentions using weekly incidence data).

(Response 3) In the preliminary ITSA analysis, we used weekly points, but because there were weeks with many zeros, we used yearly data. However, the reviewer's point is valid, and we changed to monthly data and increased the number of analysis points. Therefore, S1 Figure has been revised. We have also mentioned in the Material and Methods that the ITSA was conducted using monthly incidence. 

Material and Methods

Statistical analysis (Line 199–201)

Interrupted time series analysis (ITSA) was conducted using monthly data to determine significant changes for the incidences of individual diseases during the pre- and post-COVID-19 pandemic following the method described by Linden [30].……..

　　　　 Dengue fever 

Fig. 3: Interrupted time sires analysis conducted for monthly incidences of representative individual vector-borne diseases between the pre-pandemic and pandemic periods.

C4. It is also not clear whether the ITSAs were carried out separately for each disease or if multiple diseases were modeled as part of a multiple-group comparison. I assume that each disease was modeled separately, but line 185 refers to “multiple-group ITSA”, which would imply that the authors were also comparing the response of different diseases to the pandemic “intervention”. This section needs clarification one way or the other, and if a multiple-group analysis was conducted, the reasons for such a choice should be articulated.

(Response 4) In this study, as you noted, ITSA was carried out for individual diseases; the statement in Line 185 was my mistake. My apologies for your confusion, I have corrected "multi-group ITSA" to "single-group ITSA" (line 227). Furthermore, we have revised it as described in our reply to comment 3.

Material and Methods

Statistical analysis (Line 199-201, 208-209)

Interrupted time series analysis (ITSA) was conducted using monthly data to clear whether significant changes for the incidences of individual diseases during pre- and post-COVID-19 pandemic following the method described by Linden [30].……..……………………………………………………………………

……..Ordinary least-squares regression models designed to adjust for autocorrelation were employed for the single-group ITSA, which….

C5. In multiple sections (Abstract, Results, and Discussion) there are contrasting statements regarding whether certain diseases increased or stayed the same during the pandemic compared to the pre-pandemic period. For example, line 31-33 of the abstract says that SFTS, scrub typhus, and Japanese spotted fever remained similar and then that SFTS and scrub typhus increased significantly in the very same sentence.

(Response 5) Reply to reviewer: We have reviewed the phrase about increase or decrease of diseases throughout the entire text.

Abstract

Line 31–32: Between the pre- and post-pandemic periods, the incidence of dengue fever and malaria significantly decreased, which may related to limited human transboundary mobility (p=0.003/0.002).

Line 32–34: ….The incidence of severe fever with thrombocytopenia syndrome (SFTS), scrub typhus, and Japanese spotted fever did not show change between the two periods and either no association with human mobility.

Line 249–250: On the other hand, the incidence of relapsing fever, SFTS, scrub typhus, JSF, and Lyme disease increased during the pandemic period (+13.7%<) compared to the pre-pandemic period.

Line 270–273: Before the beginning of the COVID-19 pandemic, dengue fever/malaria onset per month did not significantly increase or decrease every month. In the first year of the pandemic (2020), there appeared to be a significant decrease in dengue fever (p=0.003) and malaria (p=0.002) onset per month,……

Line 274–275: No significant differences were observed for SFTS (p=0.265), scrub typhus (p=0.207), and JSF (p=0.514) during the pre-pandemic and pandemic periods. 

Line 328–329: We found that the onset of malaria and dengue fever showed significant decrease in the first year of the pandemic, …..

Line 330–331: On the other hand, no change in the incidence of SFTS, scrub typhus, and JSF was observed during the pre-pandemic and pandemic periods. 

Minor comments:

C1. Material and Methods

This study might benefit from the inclusion of cell-phoned derived mobility data. For example, Google has made data available on visits to multiple types of locations (https://www.google.com/covid19/mobility/). This could be an additional metric to show how movement rates changed in 2020 and 2021. In particular, one metric included by Google is visits to “parks and other natural areas”, which is relevant for several of the vector-borne diseases as mentioned in the Discussion section.

(Response 1) As suggested, we used two data sets from the COVID-19 Community Mobility Reports presented by Google, one for parks and one for homes, to show the mobility of people during the COVID-19 pandemic. In addition, epidemic curves of domestic infectious diseases were also merged.

 Domestic human mobility trends (Line 150-159)

We obtained the data sets (February 2020 to October 2022) from the COVID-19 Community Mobility Reports provided by Google [23]. These reports showed how the number of visitors to parks/outdoor spaces has changed compared to baseline days (averages from the 5‑week period from January 3, to February 6, 2020. The locations where people visited were classified into several categories. We selected parks or outdoor spaces and residential areas as locations. The reason for this selection was that parks/outdoor spaces are considered one of the most likely places to be bitten by ticks. Residential areas were also used as an indicator for staying indoors. The “parks” included places like local parks, national parks, public beaches, marinas, dog parks, plazas, and public gardens. The “residential areas” were defined as distances within a radius of 2 km from the home. The data are presented in the S5 Table.

Results

Epidemic curves of vector-borne diseases

Relationship change of domestic mobility with incidences of endemic infectious diseases 

Line 256-315: We also examined the correlation between the incidence of representative endemic infectious diseases and people’s domestic mobility change (Fig 2b). SFTS and scrub typhus showed less correlation for mobility change in parks (0.215/-0.052). JSF showed a positive correlation with mobility change in parks (0.400). However, those infectious diseases showed periodic fluctuations on an annual basis regardless of the COVID-19 pandemic.

 Figure legends

Fig. 2: Relationship change of human mobility with incidence of vector-borne diseases between January 2021 and October 2022. a) Number of foreign arrivals/Japanese returnees and incidence of vector borne diseases b) Number of visitors in parks and at home and incidence of vector-borne diseases.

References

23.Google. COVID-19 Community Mobility Reports. [Cited 2022 December 29]. Available from: https://www.google.com/covid19/mobility/

C2. Line 34 – “pan-endemic”: I assume this should be “pandemic”?

(Response 2) We have revised it as follows:

 Line 34: not show changes between the two periods.

C3. Line 154-55: Japanese encephalitis is missing from this list of domestic vector-borne diseases included in the study.

(Response 3) Thank you for your detailed review. But, this paragraph has been deleted in order to shorten the text.

C4. Line 178 – reference to Table S4 is to the incorrect table

(Response 4) Thank you for your suggestion. We deleted this sentence because shorten the text. Throughout the text, however, more attention was paid to the citation of figures and tables.

C5. Table 1 is never referenced in the body of the text.

(Response 5) Table 1 is cited in the text as follows:

Line 246–247: The incidence of VBDs among notifiable infectious diseases in our country pre- and post-pandemic is shown in Table 1. 

C6. The global incidence of several of the considered diseases, particularly Zika and chikungunya, was not stable from 2016-2019 before the COVID-19 pandemic. Zika incidence decreased from 2016-2019 as the epidemic in Latin America subsided. There should be some mention in the Discussion section that global trends might also influence incidence rates for imported diseases.

(Response 6) We have made additional mention in the discussion.

Discussion (Line 343–345)

Among the imported cases of VBDs reviewed in this study, several diseases, such as Zika and Chikungunya, had very low incidence rates during the COVID-19 pandemic period, which may not be directly linked to a decrease in the number of people entering the country [33, 34].

References

33. World Health Organization. Zika epidemiology update - February 2022. 8 February 2022

34. Pan American Health Organization. Epidemiological Update: Dengue, Chikungunya and Zika in the context of COVID-19. 23 December 2021. Available from: https://iris.paho.org/handle/10665.2/55639

C7. Lines 257-58 – This sentence is incomplete.

(Response 7) We have revised this paragraph as follow:

Line 269–277: We conducted ITSA to examine significant differences between the pre-pandemic period and the pandemic periods (Table 2, Fig 3). Before the beginning of the COVID-19 pandemic, dengue fever/malaria onset per month did not significantly increase or decrease every month. In the first year of the pandemic (2020), there appeared to be a significant decrease in dengue fever / malaria onset per month (p=0.003/0.002), followed by an insignificant increase in the annual trend of dengue fever/malaria onset per month. No significant differences were observed for SFTS, scrub typhus, and JSF during the pre-pandemic and pandemic periods (p=0.265/0.207/0.514). However, there was a change in slope from 0.021 to 0.016 for SFTS, from 0.115 to 0.511 for scrub typhus, and from 0.062 to 0.268 for JSF.

C8. Lines 268-279 – In the “Estimated infected areas of imported infectious diseases” subsection of the Results, the word “in” needs to be replaced with “from” in multiple sentences. As in “cases were reported in Asia only” should be changed to “cases were reported from Asia only”.

(Response 8) Thank you for your helpful suggestion. We have revised it as follows:

Line 285–294: We examined changes in the distribution of estimated infected areas (world regions) for VBDs between the pre-pandemic and pandemic periods (Table 3). Zika fever was reported in the pre-pandemic period, especially from Asia and the Americas, at a rate of over 40%; however, in the pandemic period, the number of cases decreased. Chikungunya fever was significantly more prevalent in Asia (91.7%) during the pre-pandemic period; however, the number of cases decreased during the pandemic period. Dengue fever was the most prevalent in Asia (87.3%) during the pre-pandemic period, and the prevalence of this disease was also common in Asia (81.1%) during the pandemic period. Malaria was especially common in Africa (75.1%) and Asia (14.2%) during the pre-pandemic period, although the number of cases decreased during the pandemic period; the ratio was generally maintained (Africa 68.1% vs. Asia 12.8%).

C9. Lines 275-277 – 81.1% of cases from Asia is not significantly more common than 87.5% of cases.

(Response 9): We have revised this sentence as follows:

Line 290–291: Dengue fever was most prevalent from Asia (87.3%) during the pre-pandemic period, and the prevalence of this disease was also common in Asia (81.1%) during the pandemic period.

C10. Table 3 – The first part of “Relapsing fever” is missing

(Response 10): Thank you for your suggestion. We have revised it in the Table 3 (Line 298)

C11. Line 322 – Malaria is spelled wrong

(Response 11): Thank you for your suggestion. We have revised it through the text.

C12. Line 372 – The start of the “Domestic infectious diseases” subsection of the Discussion discusses how SFTS, scrub typhus, and spotted fever showed no apparent decrease during the pandemic period. It is only several paragraphs later that it is mentioned that these 3 diseases, plus Lyme and relapsing fever, were actually higher during the pandemic period. It makes more sense to mention the increase first rather than simply say that they didn’t decrease.

(Response 12)Descriptions of increased incidence of Lyme disease and regression fever have been deleted. Therefore, the statement that there was no significant difference was addressed first.

Line 361–367: SFTS, scrub typhus, and JSF showed no significant changes before and after the pandemic in the analysis using ITSA. Moreover, their epidemic curves showed yearly cyclic changes during the pandemic period and did not seem to correlate with human mobility, at least not with mobility to the park. Ths suggested that domestic infectious diseases are not affected by changes in local mobility. However, a Taiwanese study using national data showed a significant decrease (52% decrease) in the incidence of scrub typhus disease in rural areas and remote islands in 2021, suggesting that a decrease in travelers to such regions may be the cause [40]. 

C13. Line 449 – Should ‘exclude’ be ‘include’?

(Response 13): We used “exclude” to mean that it could not be excluded from the analysis results, but this sentence deleted because shorten the text.

 

Additional revision by authors

1. The editorial office has pointed out that Fig. 1 may be subject to copyright. Then, the source is clearly indicated in the figure legend.

Line189–194: Fig. 1 Location of the Blakiston Line and the Itoigawa–Shizuoka Tectonic Line in the Japanese archipelago and the regional division of Japan based on these lines. The area north …… to the Itoigawa–Shizuoka Tectonic Line. This blank map was created by processing an electronic topographic map 25000 (Geospatial Information Authority of Japan) [29].

Reference

29. Geospatial Information Authority of Japan. GSI maps. [Cited 2022 December 30]. Available from: https://maps.gsi.go.jp

2. Throughout the text, the classification of imported and domestic infections has been eliminated.　This is because, for example, imported cases of domestic infectious diseases are not always zero.

3. The incidences of VBDs were presented per week, so the entire paper has been standardized to present the data per month to make it easier for the reader to read.

4. We have abbreviated vector-borne diseases as VBDs throughout the text.

5. We corrected due to change in an author's affiliation and added a coauthor for due to change in the responsible person of the department.

Title page

Line 6: ….Kazuko Yamamoto1,Jiro Fujita1, 4

Line 10: 3Center for Environmental and Health Sciences, Hokkaido University, Sapporo, Japan

Line 11:4Ohama Dai-ichi Hospital, Omoto-kai group, Naha city, Okinawa, Japan

---

## [Decision Letter · Decision Letter 1]

9 Mar 2023

PONE-D-22-18599R1Effect of non-coercive human mobility restrictions on vector-borne diseases during the COVID-19 pandemic in Japan: A descriptive epidemiological study using a national database (2016 to 2021)PLOS ONE

Dear Dr. Hibiya,

Thank you for submitting your manuscript to PLOS ONE. After careful consideration, we feel that it has merit but does not fully meet PLOS ONE’s publication criteria as it currently stands. Therefore, we invite you to submit a revised version of the manuscript that addresses the points raised during the review 

We look forward to receiving your revised manuscript.

Kind regards,

Kovy Arteaga-Livias

Academic Editor

PLOS ONE

Journal Requirements:

Reviewers' comments:

Reviewer's Responses to Questions

**Comments to the Author**

1. If the authors have adequately addressed your comments raised in a previous round of review and you feel that this manuscript is now acceptable for publication, you may indicate that here to bypass the “Comments to the Author” section, enter your conflict of interest statement in the “Confidential to Editor” section, and submit your "Accept" recommendation.

Reviewer #1: All comments have been addressed

Reviewer #2: All comments have been addressed

2. Is the manuscript technically sound, and do the data support the conclusions?

Reviewer #1: Yes

Reviewer #2: Yes

3. Has the statistical analysis been performed appropriately and rigorously? 

Reviewer #1: Yes

Reviewer #2: Yes

4. Have the authors made all data underlying the findings in their manuscript fully available?

Reviewer #1: Yes

Reviewer #2: Yes

5. Is the manuscript presented in an intelligible fashion and written in standard English?

Reviewer #1: Yes

Reviewer #2: Yes

6. Review Comments to the Author

Reviewer #1: The authors have made significant revisions to the manuscript to justify model assumptions and hypotheses. In particular, data on healthcare visits suggest that limited access may not be an important source of bias in the analysis and mobility data help to illustrate why 2020-2021 was defined as the pandemic period. The paper presents interesting analyses regarding how and why nonpharmaceutical interventions in Japan may have impacted VBD transmission. A few minor revisions, listed below, may help to provide further clarity.

Given the amount of different analyses and dataset in this paper, additional structuring and explanation to signpost the different components may be helpful. For example, the final paragraph of the introduction could contain a brief list of the research questions or parts of the manuscript. As different data sources are introduced throughout the methods section, please provide a brief explanation of why they were collected (e.g., "the number of returnees to Japan…was examined to understand how travel impacts etc"). Additionally, clarify that restriction in access to healthcare was considered (line 222) to understand whether underascertainment may have biased findings. The introduction should mention all analyses conducted, but currently does not clearly state the analysis that will be conducted on suspected region of infection.

Although the revised title is helpful to indicate moderate behavior restriction, "voluntary" may be more clear than "non-coerceive."

Line 68: The explanation provided in the discussion for potential sex differences is employment, but the hypothesis provided here involves "fear and anxiety." The suggested mechanism(s) should be consistent throughout.

Line 246: "our country" -> Japan

Line 248-50: The notation of giving the percentage with the smallest magnitude is a bit confusing. These values may be removed.

Table 1: There appear to be an errors in this table. For example, no cases of Eastern equine encephalitis were reported in either period but a 55% reduction is given, but dengue cases decreased and are listed as 0% change. Providing values for VBDs with no reported cases also appears extraneous and may make the table more difficult to read. These rows may be removed and a footnote may be used to list diseases with no reported cases.

Line 285: "infected area" -> change to "suspected region of infection" as this is more clear

Line 285-94: The presentation and interpretation of the percentages here are unclear and potentially misleading. For example, line 290-1 says "the prevalence of dengue was also common in Asia (81.1%)," which I initially read to mean that 81.1% of people in the region were infected.

Line 336: "relevant factors" -> be more specific about the eliminated factors

Line 354: "may have had a significant impact on the incident" -> can this statement be supported using evidence from Table 3?

Line 358: "current Europe" -> unclear what this means, please rephrase

Line 373: "unlikely to have been affected" -> please provide further justification for this claim

Line 407-415: The information provided regarding fever and other clinical symptoms of the different VBDs appears to be addressing whether VBDs may be misdiagnosed as Covid, but the connection is unclear. Please provide additional interpretation of these results or remove them.

Line 422: "age and sex were influenced by other factors" -> the direction of the hypothesized relationship is incorrect here.

Figure 4: The conceptual figure to indicate different covariates that may be relevant depending on the disease helps to organize the analysis, but the Venn Diagram format may be unnecessary to communicate this information. The figure could be improved by selecting a different format (perhaps a table), giving examples of specific covariates that fall under each covariate that were considered in the paper. If possible, it may also be helpful to include information about the hypothesized or estimated direction of the relationship.

Reviewer #2: The authors have addressed all the comments in previous review. My pleasure to recommend its acceptance.

7. PLOS authors have the option to publish the peer review history of their article (what does this mean?). If published, this will include your full peer review and any attached files.

Reviewer #1: No

Reviewer #2: **Yes: **Lin Wang

---

## [Author Response · Author response to Decision Letter 1]

28 Mar 2023

Review Comments to the Author 

Reviewer #1: The authors have made significant revisions to the manuscript to justify model assumptions and hypotheses. In particular, data on healthcare visits suggest that limited access may not be an important source of bias in the analysis and mobility data help to illustrate why 2020-2021 was defined as the pandemic period. The paper presents interesting analyses regarding how and why nonpharmaceutical interventions in Japan may have impacted VBD transmission. A few minor revisions, listed below, may help to provide further clarity.

Response to reviewers

Thank you for your valuable advices and comments. We have responded to each of your comments in detail.

C1. Given the amount of different analyses and dataset in this paper, additional structuring and explanation to signpost the different components may be helpful. For example, the final paragraph of the introduction could contain a brief list of the research questions or parts of the manuscript. As different data sources are introduced throughout the methods section, please provide a brief explanation of why they were collected (e.g., "the number of returnees to Japan…was examined to understand how travel impacts etc"). Additionally, clarify that restriction in access to healthcare was considered (line 222) to understand whether under ascertainment may have biased findings. The introduction should mention all analyses conducted, but currently does not clearly state the analysis that will be conducted on suspected region of infection.

R1a: We have listed the research questions and related topics in the final paragraph of the Introduction. 

➡Line 83-97

The list of the research questions and the related topics in the present study is presented below: 

1. Relationship between the incidences of VBDs and human mobility

A. Access to healthcare

B. Domestic passenger transportation activity

C. Foreign arrival

D. Returnees

E. Domestic human mobility: Long-distance travel / Population staying in local or national parks 

2. Relationship between the incidences of VBDs and commodity distribution

A. Foreign trade

B. Domestic trade

3. Relationship between the incidences of VBDs and sex and age

4. Other relevant factors with the incidence of VBDs

A. COVID-19 epidemic status in Japan

B. Geographical factor: World region / domestic region

R1b. As you have indicated, we have described why we collected the figures in the respective paragraphs in the materials and methods section.

➡Line 130

Foreign arrivals were examined to understand how foreign travel impacts the incidence of VBDs. 

➡Line 135

Returnees were examined to understand how foreign travel impacts the incidence of VBDs.

➡ Line 140-141

Cargo containers were examined to understand how overseas commodity distribution impacts the incidence of VBDs

➡Line 150-151

Domestic transport volumes were examined to determine the extent to which the long-distance movement of people and goods within a country affects the incidence of VBDs.

➡Line 177-178

Domestic human mobility was examined to understand the extent to which people's mobility in the country affected the incidence of VBDs.

➡Line 189-190

The frequency of access to healthcare was examined to understand whether underestimation may have biased findings.

 R1b. We also mentioned on the suspected regions of infection in the introduction.

➡Line 62-68

It is known that the regional incidence of VBDs often depends on the habitat range of the vector. However, it is not clear whether these differences in the regional incidence are affected by changes in human mobility due to the pandemic of COVID-19. Since the incidence of malaria and dengue fever, in which humans are the source of infection, may correlate with human mobility, a reduction in the number of travelers from endemic areas may alter the suspected transmission area [7, 8]. In addition, several tick-borne diseases are rarely transmitted by humans, so the regional differences may not change.

C2: Although the revised title is helpful to indicate moderate behavior restriction, "voluntary" may be more clear than "non-coerceive."

R2: Thank you for your appropriate recommendation. We have revised "non-coercive" to "voluntary" in the title

➡Line 1-3

Effect of voluntary human mobility restrictions on vector-borne diseases during the COVID-19 pandemic in Japan: A descriptive epidemiological study using a national database (2016 to 2021)

C3: Line 68: The explanation provided in the discussion for potential sex differences is employment, but the hypothesis provided here involves "fear and anxiety." The suggested mechanism(s) should be consistent throughout.

R3: We have revised this paragraph assuming an association between sex and age and the incidence of VBDs to be consistent within this section of the discussion.

➡Line 75-81

It is unclear whether the sex- and age-specific incidence rates of VBDs changed during the COVID-19 pandemic. A study in Taiwan on scrub typhus considered the association between age groups and specific occupations, such as farmer or soldier [12]. Furthermore, the female population in China has a reportedly higher incidence of SFTS [13], which may be due to their greater exposure to ticks during agricultural work and susceptibility to the disease after infection [13]. As a high proportion of agricultural work is known to be the work at the time of infection with VBDs in Japan [14], the sex and age distributions of affected individuals may be influenced by the occupation.

12. Lee YS, Wang PH, Tseng SJ, Ko CF, Teng HJ. Epidemiology of scrub typhus in eastern Taiwan, 2000-2004. Jpn J Infect Dis. 2006;59:235-8. Available from: https://www.niid.go.jp/niid/ja/jjid.html

13. Qian J, Wei J, Ren L, Liu Y, Feng L. Sex differences in incidence and fatality of severe fever with thrombocytopenia syndrome: a comparative study based on national surveillance data of China. J Med Virol. 2023 Mar 3. doi: 10.1002/jmv.28632. 

14. National Institute of Infectious Diseases. Scrub typhus and Japanese spotted fever in Japan 2007-2016. IASR. 2017;38: 109-112. Available from:

https://www.niid.go.jp/niid/en/basic-science/865-iasr/7342-448te.html

C4. Line 246: "our country" -> Japan

R4: We have revised “our country to Japan in the sentence.

➡Line 275-276

“The incidence of VBDs among notifiable infectious diseases during pre- and post-pandemic periods in Japan is shown in Table 1.”

C5. Line 248-50: The notation of giving the percentage with the smallest magnitude is a bit confusing. These values may be removed.

R5: We have removed the figures from the sentence as recommended.

➡Line 276-279

The incidence of Zika, Chikungunya fever, malaria, and dengue fevers, decreased by more than half during the pandemic compared to the pre-pandemic period. On the other hand, the incidence of relapsing fever, SFTS, scrub typhus, JSF, and Lyme disease increased during the pandemic compared to the pre-pandemic period.

C6. Table 1: There appear to be an errors in this table. For example, no cases of Eastern equine encephalitis were reported in either period but a 55% reduction is given, but dengue cases decreased and are listed as 0% change. Providing values for VBDs with no reported cases also appears extraneous and may make the table more difficult to read. These rows may be removed and a footnote may be used to list diseases with no reported cases.

R6: We apologize for the error which was a result of oversight. Appropriate revisions have been made and the columns that the case has not been reported has been changed to “n.r.” and it has been noted in the footnote that these diseases were not reported before or after the COVID-19 pandemic. 

➡Line 280

C7. Line 285: "infected area" -> change to "suspected region of infection" as this is more clear

R7: Thank you for your appropriate suggestion. We have revised it accordingly

➡Line 314-315

We examined changes in the distribution of suspected regions of infection (world regions) for VBDs between the pre-pandemic and pandemic periods (Table 3).

C8. Line 285-94: The presentation and interpretation of the percentages here are unclear and potentially misleading. For example, line 290-1 says "the prevalence of dengue was also common in Asia (81.1%)," which I initially read to mean that 81.1% of people in the region were infected.

R8. The text has been revised for clarity.

➡Line 314-323

We examined changes in the distribution of suspected regions of infection (world regions) for VBDs between the pre-pandemic and pandemic periods (Table 3). Zika fever was reported in the pre-pandemic period, especially by travelers from Asia and the Americas, at a rate of over 40%; however, in the pandemic period, the number of cases decreased. Chikungunya fever was significantly more prevalent in visitors from Asia (91.7%) during the pre-pandemic period; however, the number of cases decreased during the pandemic period. Dengue fever was the most prevalent among visitors from Asia during the pre-pandemic period and the pandemic periods (87.3% and 81.1%, respectively). Malaria was especially common in visitors from Africa (75.1%) and Asia (14.2%) during the pre-pandemic period. Although these cases decreased during the pandemic period, the ratio was generally maintained (Africa 68.1% vs. Asia 12.8%).

C9. Line 336: "relevant factors" -> be more specific about the eliminated factors

R9. We have indicated the specific factors that we have excluded in the cited literature.

➡ Line 364-366

The decline in dengue cases in 2020 was associated with changes in human movement behaviors, excluding the climatic and immunological factors.

C10. Line 354: "may have had a significant impact on the incident" -> can this statement be supported using evidence from Table 3?

R10. This was an oversight error. This sentence has been appropriately corrected, as can be seen in Table 3. 

➡Line 382-385

Although there was no significant decrease in the percentage of malarial infection among African visitors during the pandemic period, there was a significant decrease in the number of cases per year, from 42.25 (pre-pandemic) to 16.0 (pandemic). Thus, the decline in this number among African visitors may have a significant impact on the incidence of malaria in Japan.

C11. Line 358: "current Europe" -> unclear what this means, please rephrase

R11. The text has been revised to make it easier for readers to understand.

➡Line 389-391

It is not clear from this study whether the absence of infected travelers from Europe was due to low incidences of dengue fever, malaria and other VBDs in Europe or because the proportion of travelers was generally small.

C12. Line 373: "unlikely to have been affected" -> please provide further justification for this claim

R12. We have described the justification supporting this claim in the text.

➡Line 405-409

This claim may be supported by the fact that if tourists and others short-term visitors become infected with the tick-borne pathogen in the endemic area and, after an incubation period, develop the disease in the urban area, this will lead to an increase in the incidence in urban area. However, infection with VBDs occurred within the reporting prefecture for majority of the cases [14].

14. National Institute of Infectious Diseases. Scrub typhus and Japanese spotted fever in Japan 2007-2016. IASR. 2017;38: 109-112. Available from:

https://www.niid.go.jp/niid/en/basic-science/865-iasr/7342-448te.html

C13. Line 407-415: The information provided regarding fever and other clinical symptoms of the different VBDs appears to be addressing whether VBDs may be misdiagnosed as Covid, but the connection is unclear. Please provide additional interpretation of these results or remove them.

R13. The statement you have mentioned was added in the previous revision in response to a suggestion from an editor. However, we had deleted the sentence stating that it is difficult to differentiate symptoms of VBDs from those of COVID-19. What we really wanted to say here was that the rate of visits to medical facilities for emergency symptoms such as fever and rash was considered to be high. We have therefore added that in the revised text.

➡Line 442-449

However, the major mosquito-borne infectious diseases, dengue fever, Chikungunya fever, and Zika fever, if manifested, are characterized by fever and a generalized rash [53]. These infections are difficult to distinguish based on clinical symptoms. For tick-borne diseases, scrub typhus (n=4,185) and JSF (n=1,765), fever was observed in 95% and 99% of cases and rash in 86% and 94% of cases, respectively [54]. It is conceivable that the medical consultation rates for such emerging symptoms are not low. In addition, the majority of patients with tick-borne diseases were elderly individuals. Therefore, it is likely that the number of visits to the pediatric department was quite low, even during the non-pandemic period.

C14. Line 422: "age and sex were influenced by other factors" -> the direction of the hypothesized relationship is incorrect here.

R14. Thank you for pointing out this error. The entire paragraph has been revised, including the sentence you pointed out.

➡Line 456-459 

Host factors such as age and sex differences were found to affect tick-borne diseases, but biological, social and behavioral factors must also be taken into account to determine the true impact. Mosquito-borne diseases, such as malaria, are reportedly modified by behavioral differences between men and women [55].

55. Okiring J, Epstein A, Namuganga JF, Kamya EV, Nabende I, Nassali M, et al. Gender difference in the incidenceof malaria diagnosed at public health facilities in Uganda. Malar J. 2022;21: 22. doi: 10.1186/s12936-022-04046-4

C15. Figure 4: The conceptual figure to indicate different covariates that may be relevant depending on the disease helps to organize the analysis, but the Venn Diagram format may be unnecessary to communicate this information. The figure could be improved by selecting a different format (perhaps a table), giving examples of specific covariates that fall under each covariate that were considered in the paper. If possible, it may also be helpful to include information about the hypothesized or estimated direction of the relationship.

R15. As this is a summary of the hypotheses found in this study, we would like for it to be presented as a diagram rather than a table. The use of doughnut charts instead of Venn diagrams eliminated the intersections of each confounding factor in the Venn diagram. However, it is not possible to say from this study about the proportion of segments in the doughnut charts, so it was assumed to be set equally.

➡Line 466

Fig. 4. The hypothesized correlating factors influencing the incidence of vector-borne diseases. 

 

Reviewer #2: 

Comment: The authors have addressed all the comments in previous review. My pleasure to recommend its acceptance.

Response to reviewer: We are delighted that you have recommended acceptance to PLoS ONE. We thank you very much for your time and critical comments which helped us significantly improve our manuscript.

---

## [Decision Letter · Decision Letter 2]

16 Apr 2023

Effect of voluntary human mobility restrictions on vector-borne diseases during the COVID-19 pandemic in Japan: A descriptive epidemiological study using a national database (2016 to 2021)

PONE-D-22-18599R2

Dear Dr. Hibiya,

We’re pleased to inform you that your manuscript has been judged scientifically suitable for publication and will be formally accepted for publication once it meets all outstanding technical requirements.

Kind regards,

Kovy Arteaga-Livias

Academic Editor

PLOS ONE

Additional Editor Comments (optional):

Reviewers' comments:

Reviewer's Responses to Questions

**Comments to the Author**

1. If the authors have adequately addressed your comments raised in a previous round of review and you feel that this manuscript is now acceptable for publication, you may indicate that here to bypass the “Comments to the Author” section, enter your conflict of interest statement in the “Confidential to Editor” section, and submit your "Accept" recommendation.

Reviewer #1: All comments have been addressed

2. Is the manuscript technically sound, and do the data support the conclusions?

Reviewer #1: Yes

3. Has the statistical analysis been performed appropriately and rigorously? 

Reviewer #1: Yes

4. Have the authors made all data underlying the findings in their manuscript fully available?

Reviewer #1: Yes

5. Is the manuscript presented in an intelligible fashion and written in standard English?

Reviewer #1: Yes

6. Review Comments to the Author

Reviewer #1: The authors have done an excellent job addressing reviewer questions. I recommend its acceptance for publication.

Two small additional recommendations for clarity are provided here. Please ensure that the "not reported" labeled is applied consistently in Table 1 (epidemic typhus and tularemia should receive this label). The table could also be sorted by the % change per year column so that rows with cases reported are grouped together, making it easier for the reader to focus on potential diseases of interest. The language about changes in distribution of suspected regions of infection (lines 314-323) could also be further clarified. Rather than making statements about the relative prevalence in different regions, this section is actually comparing the relative share of total imported cases suspected to be attributable to travels from a given region.

7. PLOS authors have the option to publish the peer review history of their article (what does this mean?). If published, this will include your full peer review and any attached files.

Reviewer #1: No

---

## [Editor Report · Acceptance letter]

16 May 2023

PONE-D-22-18599R2 

Effect of voluntary human mobility restrictions on vector-borne diseases during the COVID-19 pandemic in Japan: A descriptive epidemiological study using a national database (2016 to 2021) 

Dear Dr. Hibiya:

I'm pleased to inform you that your manuscript has been deemed suitable for publication in PLOS ONE. Congratulations! Your manuscript is now with our production department. 

Kind regards, 

on behalf of

Dr. Kovy Arteaga-Livias 

Academic Editor

PLOS ONE